# Bayesian Estimation for the Difference between Coefficients of Quartile Variation of Delta-Lognormal Distributions: An Application to Rainfall in Thailand

**Noppadon Yosboonruang** [1] and **Sa-Aat Niwitpong** [2,*]

1   Department of Statistics, School of Science, University of Phayao, Phayao 56000, Thailand; noppadon.yo@up.ac.th

2   Department of Applied Statistics, Faculty of Applied Science, King Mongkut's University of Technology North Bangkok, Bangkok 10800, Thailand

\*   Correspondence: sa-aat.n@sci.kmutnb.ac.th

**Abstract:** The coefficient of quartile variation is a valuable measure used to assess data dispersion when it deviates from a normal distribution or displays skewness. In this study, we focus specifically on the delta-lognormal distribution. The lognormal distribution is characterized by its asymmetrical nature and comprises exclusively positive values. However, when these values undergo a logarithmic transformation, they conform to a symmetrical (or normal) distribution. Consequently, this research aims to establish confidence intervals for the difference between coefficients of quartile variation within lognormal distributions incorporating zero values. We employ the Bayesian, generalized confidence interval, and fiducial generalized confidence interval methods to construct these intervals, involving data simulation using RStudio software. We evaluate the performance of these methods based on coverage probabilities and average lengths. Our findings indicate that the Bayesian method, employing Jeffreys' prior, performs well in low variability, while the generalized confidence interval method is more suitable for higher variability. Therefore, we recommend using both approaches to construct confidence intervals for the difference between the coefficients of the quartile variation in lognormal distributions that include zero values. Furthermore, we apply these methods to rainfall data in Thailand to illustrate their alignment with actual and simulated data.

**Keywords:** delta-lognormal distribution; coefficient of quartile variation; Bayesian; generalized confidence interval; rainfall

## 1. Introduction

Climate change significantly impacts weather patterns worldwide, including in Thailand, where changes in rainfall dispersion have become particularly noticeable [1]. Thailand has a tropical climate characterized by a rainy season from May to October and a dry season from November to April [2]. However, recent years have witnessed irregular and extreme weather conditions, causing disruptions to traditional rainfall patterns. Several researchers have investigated rainfall dispersion in Thailand, such as Kumphon et al. [3], Szyniszewska and Waylen [4], and Thodsan et al. [5]. Additionally, studies on statistical inference for rainfall distribution in Thailand have been reported by Maneerat et al. [6], Khooriphan et al. [7], Yosboonruang et al. [8,9], and Thangjai et al. [10]. Previous research on statistical inference has primarily focused on measuring rainfall dispersion using variance and coefficient of variation. However, when rainfall dispersion is highly skewed, the coefficient of quartile variation (CQV) becomes a more appropriate tool for analyzing this data type.

The CQV, which stands for the quartile coefficient of dispersion, is a statistical measure that assesses the relative dispersion or variability within a dataset. It quantifies the spread of the data concerning its central tendency, represented by the median. A higher coefficient indicates a more significant variability or dispersion in the dataset, indicating that the

values are more spread out from the median. Conversely, a lower coefficient suggests less variability, indicating that the values are closer and more tightly clustered around the median. When the data follow a non-normal distribution, the CQV performs better than the coefficient of variation in measuring relative dispersion. Additionally, when outlier values are present, the CQV is a more appropriate measure for quantifying data dispersion [11]. The CQV has been applied in various subject areas. For instance, Hussein and Morgan [12] used the CQV to measure intravertebral density heterogeneity. Marcoulaki et al. [13] employed the CQV to assess dispersion in computer simulation data for designing central pipeline systems. Chatterjee et al. [14] compared land surface temperature and radiant temperature images at notable coal fire locations using the CQV. Antonetti et al. [15] incorporated water temperature simulations into a fish habitat model and measured thermal heterogeneity using the CQV. Furthermore, researchers have examined statistical inferences regarding the quartile coefficient of variation. Bonett [16] constructed confidence intervals for the CQV, which applied to normal and non-normal distributions. Ambati et al. [17] introduced ratio- and regression-type estimators for estimating the CQV in finite populations. Javed et al. [18] proposed a class of ratio estimators for estimating population variance, utilizing the CQV of an auxiliary variable. Altunkaynak and Gamgam [19] recommended the bootstrap method to establish confidence intervals for the CQV in non-normal distributions. Singh et al. [20] and Ahmed and Shabbir [21] identified an error in the mean squared error of Ambati et al. [17] and rectified it using auxiliary information. Ahmed and Shabbir [21] also presented the Rao regression-type estimator for estimating the CQV with an auxiliary variable. In 2022, Eppen et al. [22] proposed naïve, Rao, and regression estimators for estimating the CQV with an auxiliary variable. Singh and Usman [23] expanded upon the methods introduced by Ambati et al. [17] to estimate the CQV for missing data. Furthermore, Yosboonruang et al. [24] developed a confidence interval for the CQV of a zero-inflated lognormal distribution.

The lognormal distribution plays a significant role in climate change studies due to its ability to model skewed data, analyze extreme events, quantify uncertainty, and facilitate econometric analyses. According to the rainfall data, several researchers have reported that the data follow a lognormal distribution with zero values, also known as a delta-lognormal distribution [8,9,25–28]. The delta-lognormal distribution consists of positive values following a lognormal distribution and actual zero values following a binomial distribution. The lognormal distribution is asymmetrical in shape. Nevertheless, if the values of the lognormal random variable undergo a logarithmic transformation, they conform to a symmetrical distribution, commonly known as the normal distribution. This distribution has attracted significant interest from researchers studying statistical inference related to it. For example, Li et al. [29] presented generalized and fiducial inference approaches for estimating the mean of a lognormal distribution with excess zeros, with the fiducial approach demonstrating superior performance. Wu and Hsieh [30] constructed a generalized confidence interval (GCI) for the mean of the delta-lognormal distribution using an asymptotic generalized pivotal quantity (GPQ), and their method showed excellent performance. Hasan and Krishnamoorthy [31] introduced fiducial confidence intervals and the method of variance estimate recovery (MOVER) to estimate the mean of a delta-lognormal distribution, receiving recognition for the effectiveness of their proposed methods. In their 2022 study, Zhang et al. [32] proposed the fiducial generalized pivotal quantity (FGPQ) and employed MOVER with FGPQ to construct simultaneous confidence intervals for ratios of means in zero-inflated lognormal distributions, which were also highly recognized for their effectiveness. Furthermore, Yosboonruang et al. [9] introduced various methods to estimate the ratio of coefficients of variation of lognormal distributions with excess zeros, including the fiducial generalized confidence interval (FGCI), Bayesian methods, and the Wald and Fieller log-likelihood methods, with the Bayesian method proving to be the most effective. Recently, in 2023, Thangjai et al. [10] employed the FGCI, Bayesian, and bootstrap methods to establish confidence intervals for the ratio of percentiles of delta-lognormal distributions, with the Bayesian method demonstrating superior performance. Various

studies have addressed estimating parameters for lognormal distributions with excess zero values. In this article, we focus on estimating the dispersion of a dataset that follows a delta-lognormal distribution. One effective method for estimating this dispersion is the CQV, building upon the methodology proposed by Yosboonruang and Niwitpong [24] to examine and compare the dispersion between two datasets. Specifically, we aim to construct the highest posterior density (HPD) and confidence intervals for the difference between the CQVs of two delta-lognormal distributions.

The following section presents Bayesian approaches utilizing multiple priors, GCI, and FGCI to construct HPD and confidence intervals. Section 3 provides the simulation results and an empirical study. The final section encompasses the discussion and conclusions of the study.

## 2. Materials and Methods

Let $Y_{ij}$, $i = 1, 2$, $j = 1, 2, \ldots, n_i$ be random variables from $n$ observations of delta-lognormal distributions denoted by $Y_{ij} \sim \Delta(\mu_i, \sigma_i^2, \delta_{i_0})$, where $\mu_i$, $\sigma_i^2$, and $\delta_{i_0}$ represent the mean, variance, and probability of zero values, respectively. For $Y_{ij} > 0$, $X_{ij} = \ln(Y_{ij}) \sim N(\mu_i, \sigma_i^2)$ follows a lognormal distribution while $Y_{ij} = 0$ follows a binomial distribution. Let $n_{i_0}$ and $n_{i_1}$ be the numbers of zero and positive values, respectively, such that $n_i = n_{i_0} + n_{i_1}$. Aitchison [33] derived the mean and variance of $Y_{ij}$ as $\mu_Y = \delta_{i_1} \exp(\mu_i + \sigma_i^2/2)$ and $\sigma_Y^2 = \delta_{i_1} \exp(2\mu_i + \sigma_i^2)[\exp(\sigma_i^2) - \delta_{i_1}]$, where $\delta_{i_1}$ is the probability of positive values.

The CQV is a descriptive statistic used to measure the dispersion between data sets that have different units or to compare within data sets that have different mean values. The CQV is defined by the first and third quartiles as follows:

$$\varphi_i = \frac{Q_{3_i} - Q_{1_i}}{Q_{3_i} + Q_{1_i}}, \tag{1}$$

where $Q_{1_i}$ and $Q_{3_i}$ denote the first and third quartiles of $Y_{ij}$, respectively. The quartiles are determined according to Hasan and Krishnamoorthy [31] as

$$Q_{r_i} = \exp\left\{\mu_i + \Phi^{-1}\left[\frac{\frac{r_i}{4} - (1 - \delta_{i_1})}{\delta_{i_1}}\right]\sigma_i\right\}, \ \delta_{i_1} > 1 - \frac{r_i}{4}, \tag{2}$$

where $\Phi$ is the cumulative standard normal distribution. Since this study focuses on the difference between CQVs, it is defined as

$$\Psi = \varphi_1 - \varphi_2. \tag{3}$$

This study introduces Bayesian and GCI methods to establish HPD and confidence intervals for the difference between CQVs.

### 2.1. Bayesian Method

Nowadays, research on statistical inferences and applications is focused on the Bayesian approach because it relies on the population distribution to estimate the parameter of interest [34]. In Bayesian inference, the parameters of interest are directly illustrated by the probability distribution, which is defined as random variables [35].

Regarding the unknown parameters of the delta-lognormal distributions, namely $\delta_{i_0}$, $\mu_i$, and $\sigma_i^2$ where $\delta_{i_0} = 1 - \delta_{i_1}$, the joint likelihood function can be defined as

$$L\left(\delta_{i_0}, \mu_i, \sigma_i^2 \mid \mathbf{y}_{ij}\right) \propto \prod_{i=1}^{2}\left\{\delta_{i_0}^{n_{i_0}}\left(1 - \delta_{i_0}\right)^{n_{i_1}}\prod_{j=1}^{n_{i_1}}\frac{1}{\sigma_i}\exp\left[-\frac{1}{2\sigma_i^2}\left(\ln\left(y_{ij}\right) - \mu_i\right)^2\right]\right\}. \tag{4}$$

Using Equation (4), the Fisher information matrix of parameters $\theta = \left(\delta_{1_0}, \mu_1, \sigma_1^2, \delta_{2_0}, \mu_2, \sigma_2^2\right)$ can be derived by taking the second-order derivative of the log-likelihood function:

$$I(\theta) = \text{diag}\left[\frac{n_1}{\delta_{1_0}\left(1-\delta_{1_0}\right)} \quad \frac{n_{1_1}}{\sigma_1^2} \quad \frac{n_1\left(1-\delta_{1_0}\right)}{2\left(\sigma_1^2\right)^2} \quad \frac{n_2}{\delta_{2_0}\left(1-\delta_{2_0}\right)} \quad \frac{n_{2_1}}{\sigma_2^2} \quad \frac{n_2\left(1-\delta_{2_0}\right)}{2\left(\sigma_2^2\right)^2}\right]. \tag{5}$$

To estimate the difference between CQVs, HPD intervals are constructed based on the posterior distribution, which is updated using the concept of Bayes' theorem defined as

$$p\left(\theta_i|y_{i1}, y_{i2}, ..., y_{in_i}\right) = \frac{p(\theta_i)p\left(y_{i1}, y_{i2}, ..., y_{in_i}|\theta_i\right)}{p\left(y_{i1}, y_{i2}, ..., y_{in_i}\right)}.$$

Since the parameters of interest in this study are $\delta_{i_0}$, $\mu_i$, and $\sigma_i^2$, the posterior of these parameters is computed by integrating the likelihood function in Equation (4) with the prior density function for a delta-lognormal distribution, $p\left(\delta_{i_0}, \mu_i, \sigma_i^2|y_{ij}\right)$. Therefore, the posterior density of $\delta_{i_0}$, $\mu_i$ and $\sigma_i^2$ can be derived as follows:

$$p\left(\delta_{i_0}|y_{ij}\right) = \iint p\left(\delta_{i_0}, \mu_i, \sigma_i^2|y_{ij}\right)d\mu_i d\sigma_i^2, \tag{6}$$

$$p\left(\mu_i|\sigma_i^2, y_{ij}\right) = \int p\left(\delta_{i_0}, \mu_i, \sigma_i^2|y_{ij}\right)d\delta_{i_0}, \tag{7}$$

and

$$p\left(\sigma_i^2|y_{ij}\right) = \iint p\left(\delta_{i_0}, \mu_i, \sigma_i^2|y_{ij}\right)d\delta_{i_0}d\mu_i, \tag{8}$$

respectively.

Furthermore, Bayesian deep learning can also serve as an alternative approach for generating posterior distributions. By incorporating Bayesian inference techniques, this approach provides several advantages, including robust uncertainty estimation and a principled approach to mitigating overfitting issues. Estimating posterior distributions allows a more comprehensive understanding of the uncertainty associated with the model's parameters, given the observed data. For a more in-depth exploration of the Bayesian deep learning method, we recommend referring to the research conducted by Zhuang et al. [36].

This article selected three prior distributions, namely the normal gamma prior, Jeffrey's prior, and the uniform prior for the Bayesian method because these prior distributions can return the closed-form solutions of the posterior distributions. But whenever choosing priors in other forms, the posterior distributions may not have closed-form solutions or follow regular distributions. Therefore, it is necessary to find alternative inference methods, such as Bayesian variable sampling or Markov chain Monte Carlo sampling (MCMC), etc. [37].

### 2.1.1. The Normal-Gamma Prior

Choosing hyperparameters in the normal-gamma prior involves determining appropriate values for the mean and precision parameters of the normal distribution and the shape and rate parameters of the gamma distribution. These hyperparameters play a crucial role in shaping the prior distribution and subsequently influence the posterior distribution in Bayesian inference.

Maneerat et al. [6] utilized the conjugate families proposed by DeGroot [38] for a normal random sample to derive the posterior distribution of parameters in the normal-gamma prior. The posterior distributions of $\delta_{i_0}$, $\mu_i$, and $\sigma_i^2$ are as follows: $p\left(\delta_{i_0}|y_{ij}\right) \sim Beta\left(n_{i_0} + d_i, n_{i_1} + d_i\right)$, where $d_i = \frac{1}{6}\left(2 + z_{\frac{\alpha}{2}}^2\right)$; $p\left(\mu_i|\sigma_i^2, y_{ij}\right) \sim t_{n_{i_1}-1}\left(\hat{\mu}_i, \frac{\frac{1}{2}\sum_{i=1}^{n_{i_1}}\left(\ln\left(y_{ij}\right)-\hat{\mu}_i\right)^2}{\left(\frac{n_{i_1}-1}{2}\right)n_{i_1}}\right)$; and $p\left(\sigma_i^2|y_{ij}\right) \sim Inv - Gamma\left[\left(n_{i_1} - 1\right)/2, \frac{1}{2}\sum_{i=1}^{n_{i_1}}\left(\ln\left(y_{ij}\right) - \hat{\mu}_i\right)^2\right]$, respectively.

2.1.2. Jeffreys' Prior

According to the delta-lognormal distribution, which is a mixture of the lognormal and binomial distributions, the parameters of interest are $\delta_{i_0}$, $\mu_i$, and $\sigma_i^2$. Following the concept of Jeffreys [39], the prior distributions for these parameters can be obtained by taking the square root of the determinant of the Fisher information matrix (Equation (5)), resulting in $p(\theta_i) = \sqrt{|I(\theta)|}$. Specifically, the Jeffreys prior for $\delta_{i_0}$ is $p(\delta_{i_0}) = (\delta_{i_0}\delta_{i_1})^{-\frac{1}{2}}$. For the lognormal distribution, the prior distributions for $\mu_i$ and $\sigma_i^2$ are $p(\mu_i) \propto \exp\left\{ -\frac{n_{i_1}}{2\sigma_i^2}(\mu_i - \hat{\mu}_i)^2 \right\}$ and $p(\sigma_i^2) = \sigma_i^{-2}$ [40]. Consequently, the posterior distributions of these parameters can be computed using Equations (6) - (8) as follows: $p(\delta_{i_0}|y_{ij}) \sim Beta(n_{i_0} + 0.5, n_{i_1} + 0.5)$, $p(\mu_i|\sigma_i^2, y_{ij}) \sim N(\hat{\mu}_i, \sigma_i^2/n_{i_1})$, and $p(\sigma_i^2|y_{ij}) \sim Inv - Gamma\left[(n_{i_1} - 1)/2, (n_{i_1} - 1)\hat{\sigma}_i^2/2\right]$.

2.1.3. The Uniform Prior

According to the uniform prior, it represents a constant function of an a priori probability that assigns equal probabilities to all possible values [41,42]. Therefore, the uniform priors for the parameters of the binomial and lognormal distributions are proportional to 1. By integrating the prior density function for a delta-lognormal distribution, the posterior distributions of $\delta_{i_0}$, $\mu_i$, and $\sigma_i^2$ can be determined as follows: $p(\delta_{i_0}|y_{ij}) \sim Beta(n_{i_0} + 1, n_{i_1} + 1)$, $p(\mu_i|\sigma_i^2, y_{ij}) \sim N(\hat{\mu}_i, \sigma_i^2/n_{i_1})$, and $p(\sigma_i^2|y_{ij}) \sim Inv - Gamma\left[(n_{i_1} - 2)/2, (n_{i_1} - 2)\hat{\sigma}_i^2/2\right]$.

Using the posteriors of $\delta_{i_0}$, $\mu_i$, and $\sigma_i^2$ obtained for each prior, we substitute these posterior distribution into Equation (2) to calculate the difference between CQVs. Subsequently, we construct the HPD intervals for all methods using the HDInterval package in the R statistical program, following the outlined Algorithm 1 below.

---

**Algorithm 1** Steps to construct HPD intervals for the Bayesian method.

---

**Step 1.** Generate $Y_{ij}$, where $i = 1, 2$ and $j = 1, 2, ..., n_i$, from the delta-lognormal distributions.
**Step 2.** Compute $\hat{\delta}_{i_0}$, $\hat{\mu}_i$, and $\hat{\sigma}_i^2$.
**Step 3.** Generate the posterior densities of $\delta_{i_0}$, $\mu_i$, and $\sigma_i^2$ using each prior.
**Step 4.** Compute $Q_{r_i}$ using Equation (2).
**Step 5.** Compute $\varphi_i$ using Equation (1).
**Step 6.** Compute $\Psi$ using Equation (3).
**Step 7.** Repeat Steps 3–6 for a total of 2000 times.
**Step 8.** Construct HPD intervals for $\Psi$ using each prior.
**Step 9.** Repeat Steps 1–8 for a total of 10,000 times.

---

2.2. *Generalized Confidence Interval*

The concept of the GCI was introduced by Weerahandi [43]. It is based on the GPQs of the parameters of interest. Furthermore, the construction of confidence intervals for the model parameters using the generalized inference method is discussed in [44]. In this context, the random variables $Y_{ij}$, where $i = 1, 2$ and $j = 1, 2, ..., n_i$, follow delta-lognormal distributions. Referring to Equations (1)–(3), the parameters of interest are $\delta_{i_1}$, $\mu_i$, and $\sigma_i$. Let $y_{ij}$, where $i = 1, 2$ and $j = 1, 2, ..., n_i$ are the observed values of the random variables $Y_{ij}$. The GPQs for these parameters possess two important properties: (1) the distribution of GPQs is free from all unknown parameters, and (2) the observed values of GPQs do not depend on the nuisance parameter. Following Wu and Hsieh [30], they computed the variance stabilizing transformation of a binomial distribution using the concept of DasGupta [45]. The GPQ for $\delta_{i_1}$ is given by

$$R_{\delta_{i_1}}^{gci} = \sin^2\left[\arcsin\sqrt{\hat{\delta}_{i_1}} - \frac{1}{2\sqrt{n_i}}Z_i\right], \tag{9}$$

where $Z_i \sim N(0,1)$. Moreover, they used the idea of Krishnamoorthy and Mathew [46] to compute the GPQs for $\mu_i$ and $\sigma_i^2$ as follows:

$$R_{\mu_i} = \hat{\mu}_i - Z_i \sqrt{\frac{(n_{i_1} - 1)\hat{\sigma}_i^2}{n_{i_1} \chi_{n_{i_1}-1}^2}}, \tag{10}$$

where $Z_i \sim N(0,1)$, and

$$R_{\sigma_i} = \sqrt{\frac{(n_{i_1} - 1)\hat{\sigma}_i^2}{\chi_{n_{i_1}-1}^2}}, \tag{11}$$

The pivotal quantities $R_{\delta_{i_1}}^{gci}$, $R_{\mu_i}$, and $R_{\sigma_i}$ are consistent with the properties of GPQs. Therefore, we can express $R_{Q_{r_i}}^{gci}$ as follows:

$$R_{Q_{r_i}}^{gci} = \exp\left\{ R_{\mu_i} + \Phi^{-1}\left[ \frac{\frac{r_i}{4} - \left(1 - R_{\delta_{i_1}}^{gci}\right)}{R_{\delta_{i_1}}^{gci}} \right] R_{\sigma_i} \right\}, R_{\delta_{i_1}}^{gci} > 1 - \frac{R_{\delta_{i_1}}^{gci}}{4}. \tag{12}$$

By substituting the pivotal quantity from Equation (12) into Equation (1), we obtain

$$R_{\varphi_i}^{gci} = \frac{R_{Q3_i}^{gci} - R_{Q1_i}^{gci}}{R_{Q3_i}^{gci} + R_{Q1_i}^{gci}}. \tag{13}$$

Hence, the pivotal quantity for the difference between CQVs is given by

$$R_\Psi^{gci} = R_{\varphi_1}^{gci} - R_{\varphi_2}^{gci}. \tag{14}$$

Consequently, the $(1-\alpha)100\%$ confidence interval for $\Psi$ can be expressed as

$$CI_\Psi^{gci} = \left[ R_\Psi^{gci}(\alpha/2), R_\Psi^{gci}(1-\alpha/2) \right], \tag{15}$$

where $R_\Psi^{gci}(\alpha/2)$ and $R_\Psi^{gci}(1-\alpha/2)$ represent the $100(\alpha/2)$-th and $100(1-\alpha/2)$-th percentiles of $R_\Psi$, respectively. The steps for constructing confidence intervals using the GCI method are presented in Algorithm 2.

---

**Algorithm 2** Steps to construct confidence interval for the GCI method.

---

**Step 1.** Generate $Y_{ij}$, where $i = 1,2$ and $j = 1,2,...,n_i$, from the delta-lognormal distributions.
**Step 2.** Compute the estimates $\hat{\delta}_{i_1}$, $\hat{\mu}_i$, and $\hat{\sigma}_i^2$.
**Step 3.** Generate random variables $Z_i \sim N(0,1)$ and $\chi_{n_{i_1}-1}^2$.
**Step 4.** Compute the pivotal quantities $R_{\delta_{i_1}}^{gci}$, $R_{\mu_i}$, and $R_{\sigma_i}$.
**Step 5.** Repeat Steps 3–4 for a total of 2000 times.
**Step 6.** Construct the confidence interval for $\Psi$.
**Step 7.** Repeat Steps 1–6 for a total of 10,000 times.

---

### 2.3. Fiducial Generalized Confidence Interval

We extended the FGCI method, as proposed by Yosboonruang et al. [24], to handle the construction of the confidence interval for the difference between CQVs in the delta-lognormal distribution. Consequently, the parameters of interest, based on Equation (2), are $\delta_{i_1}$, $\mu_i$, and $\sigma_i$. The fiducial quantities for these parameters can be expressed as $R_{\delta_{i_1}}^{fgci} \sim Beta\left(n_i - n_{i_1}, n_{i_1} + 0.5\right)$, while $R_{\mu_i}$ and $R_\sigma$ are represented by Equations (10) and (11), re-

spectively. Furthermore, by utilizing Equation (2), we can determine the fiducial quantity for $Q_{r_i}$ as follows:

$$R_{Q_{r_i}}^{fgci} = \exp\left\{ R_{\mu_i} + \Phi^{-1}\left[ \frac{\frac{r_i}{4} - \left(1 - R_{\delta_{i_1}}^{fgci}\right)}{R_{\delta_{i_1}}^{fgci}} \right] R_{\sigma_i} \right\}, R_{\delta_{i_1}}^{fgci} > 1 - \frac{R_{\delta_{i_1}}^{fgci}}{4}. \tag{16}$$

Substituting $R_{Q_{r_i}}^{fgci}$ into Equation (1), we can obtain the fiducial quantity for $\varphi_i$:

$$R_{\varphi_i}^{fgci} = \frac{R_{Q_{3_i}}^{fgci} - R_{Q_{1_i}}^{fgci}}{R_{Q_{3_i}}^{fgci} + R_{Q_{1_i}}^{fgci}}. \tag{17}$$

Accordingly, the fiducial quantity for the difference between CQVs, denoted as $R_{\Psi}^{fgci}$, can be represented as

$$R_{\Psi}^{fgci} = R_{\varphi_1}^{fgci} - R_{\varphi_2}^{fgci}. \tag{18}$$

Consequently, the $(1 - \alpha)100\%$ confidence interval for $\Psi$ can be expressed as

$$CI_{\Psi}^{fgci} = \left[ R_{\Psi}^{fgci}(\alpha/2), R_{\Psi}^{fgci}(1 - \alpha/2) \right], \tag{19}$$

where $R_{\Psi}^{fgci}(\alpha/2)$ and $R_{\Psi}^{fgci}(1 - \alpha/2)$ represent the $100(\alpha/2)$-th and $100(1 - \alpha/2)$-th percentiles of $R_{\Psi}$, respectively. Algorithm 3 outlines the procedure for constructing confidence intervals using the FGCI method.

---

**Algorithm 3** Steps to construct confidence interval for the FGCI method.

---

**Step 1.** Generate $Y_{ij}$, where $i = 1, 2$ and $j = 1, 2, ..., n_i$, from the delta-lognormal distributions.
**Step 2.** Compute the estimates $\hat{\delta}_{i_1}$, $\hat{\mu}_i$, and $\hat{\sigma}_i^2$.
**Step 3.** Generate random variables $Z_i \sim N(0, 1)$ and $\chi_{n_{i_1}-1}^2$.
**Step 4.** Compute the pivotal quantities $R_{\delta_{i_1}}^{fgci}$, $R_{\mu_i}$, and $R_{\sigma_i}$.
**Step 5.** Repeat Steps 3–4 for a total of 2000 times.
**Step 6.** Construct the confidence interval for $\Psi$.
**Step 7.** Repeat Steps 1–6 for a total of 10,000 times.

---

## 3. Results

### 3.1. Simulation Study

To compare the performance of various methods, including Bayesian methods based on three different priors (normal-gamma prior—B.NG, Jeffreys' prior—B.J, and uniform prior—B.U), as well as the GCI and FGCI methods, we conducted simulation studies using Monte Carlo simulation in RStudio version 2022.12.0+353. The goal was to compute the coverage probabilities and average lengths of the HPD and confidence intervals. A favorable method would have coverage probabilities close to or greater than the nominal confidence level of 0.95 and the narrowest intervals. The simulation settings were as follows: sample sizes of 5, 10, 15, 30, 50, 100 and 200; mean values following Hasan and Krishnamoorthy [31] as $-\sigma^2/2$; probabilities of positive values as 0.80, 0.85, 0.90, and 0.95 (aligned with Equation (2)); and population variances of 1, 2, 3, 5, and 10. We performed 10,000 replicates for all parameter combinations and 2000 replicates for the Bayesian and GCI methods.

The results in Tables 1 and 2 provide the HPD and confidence intervals for the difference between CQVs. In cases with small sample sizes and low variance of all probabilities of positive values using the B.NG and B.J methods, the coverage probabilities fall below the nominal confidence level of 0.95. However, the coverage probabilities are either close to or exceed 0.95 for other cases. Figure 1 displays the coverage probabilities of GCI for cases

with equal sample sizes. It consistently demonstrates coverage probabilities around 0.95, which are lower than the probabilities obtained from other methods when the sample sizes exceed 5, irrespective of the probabilities of positive values. However, for a small sample size of 5, the B.J method outperforms the other methods.

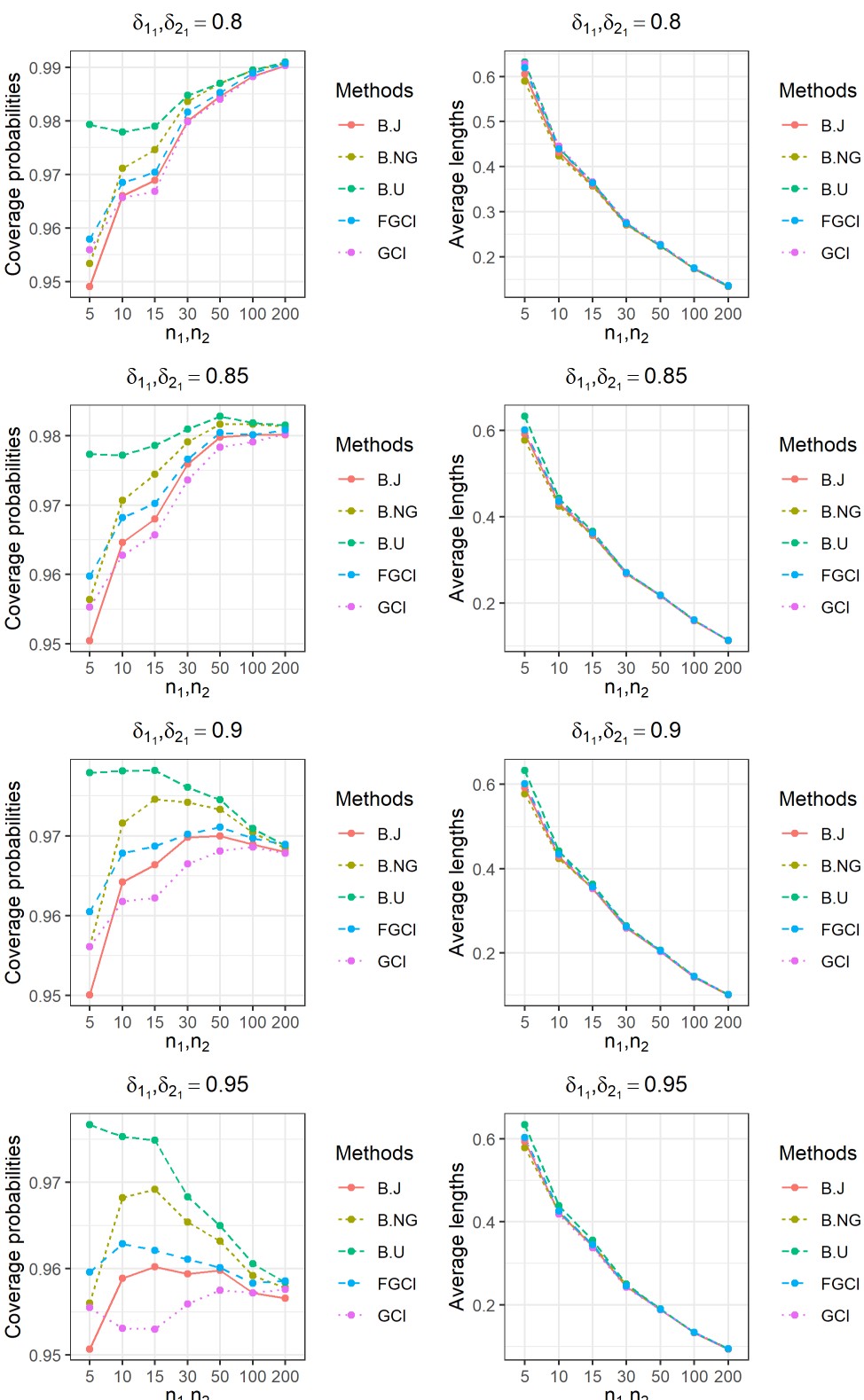

**Figure 1.** Comparison of the coverage probabilities and average lengths of proposed methods for cases with varying sample sizes and probabilities of positive values, assuming equal sample sizes.

Moving on to Figure 2, which depicts cases with unequal sample sizes, reveals that the coverage probabilities of GCI are either close to or higher than 0.95 for at least one sample size larger than 5, accompanied by probabilities equal to 0.8 or 0.85. Moreover, as the sample size increases and the probability of positive values reaches or exceeds 0.90, the performance of B.J becomes comparable to or better than that of GCI.

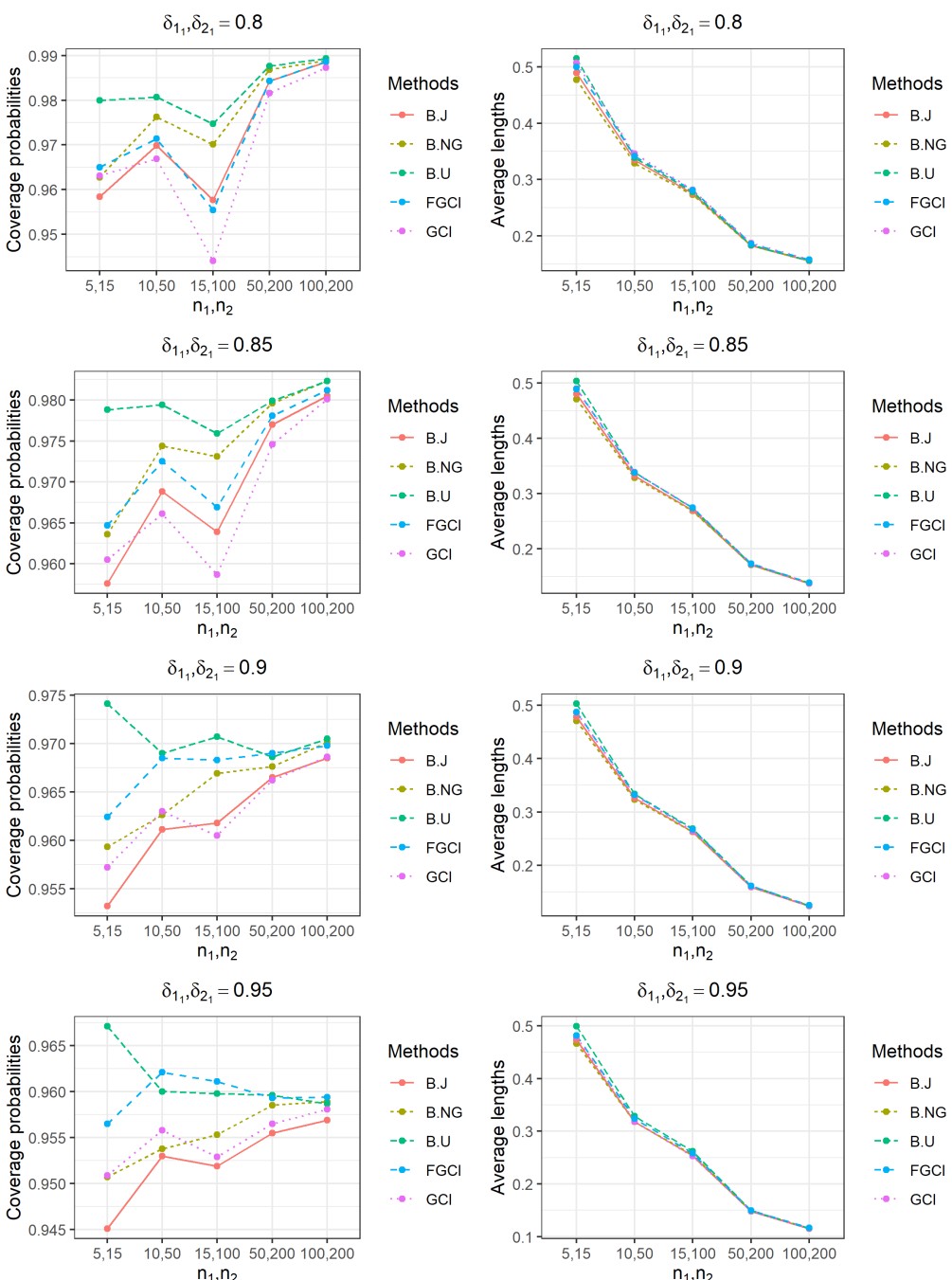

**Figure 2.** Comparison the coverage probabilities and average lengths of proposed methods for cases with varying sample sizes and probabilities of positive values, assuming unequal sample sizes.

**Table 1.** The coverage probabilities and average lengths of two-sided 95% HPD and confidence intervals for the difference between CQVs in delta-lognormal distributions are presented for cases with equal sample sizes.

| $n_1, n_2$ | $\delta_1, \delta_2$ | $\sigma_1^2, \sigma_2^2$ | B.NG | | B.J | | B.U | | GCI | | FGCI | |
|---|---|---|---|---|---|---|---|---|---|---|---|---|
| | | | CP | AL | CP | AL | CP | AL | CP | AL | CP | AL |
| 5, 5 | 0.80, 0.80 | 1, 1 | 0.9350 | 0.8378 | 0.9306 | 0.8452 | 0.9693 | 0.9323 | 0.9589 | 0.8671 | 0.9605 | **0.8649** |
| | | 2, 2 | 0.9405 | 0.7103 | 0.9355 | 0.7250 | 0.9725 | **0.6698** | 0.9550 | 0.7480 | 0.9578 | 0.7408 |
| | | 3, 3 | 0.9466 | 0.6136 | 0.9428 | 0.6321 | 0.9768 | 0.6797 | 0.9553 | 0.6546 | 0.9570 | **0.6449** |
| | | 5, 5 | 0.9639 | **0.4840** | 0.9577 | 0.5048 | 0.9852 | 0.5387 | 0.9539 | 0.5267 | 0.9563 | 0.5157 |
| | | 10, 10 | 0.9808 | **0.3029** | 0.9787 | 0.3224 | 0.9929 | 0.3419 | 0.9565 | 0.3439 | 0.9577 | 0.3328 |
| | 0.85, 0.85 | 1, 1 | 0.9381 | 0.8147 | 0.9331 | 0.8154 | 0.9671 | 0.8959 | 0.9558 | **0.8196** | 0.9606 | 0.8292 |
| | | 2, 2 | 0.9437 | 0.6965 | 0.9381 | 0.7082 | 0.9677 | 0.7603 | 0.9549 | **0.7170** | 0.9594 | 0.7192 |
| | | 3, 3 | 0.9505 | **0.6047** | 0.9438 | 0.6214 | 0.9751 | 0.6600 | 0.9533 | 0.6315 | 0.9581 | 0.6305 |
| | | 5, 5 | 0.9662 | **0.4740** | 0.9586 | 0.4952 | 0.9839 | 0.5194 | 0.9569 | 0.5063 | 0.9605 | 0.5019 |
| | | 10, 10 | 0.9835 | **0.2963** | 0.9791 | 0.3176 | 0.9928 | 0.3286 | 0.9558 | 0.3306 | 0.9605 | 0.3247 |
| | 0.90, 0.90 | 1, 1 | 0.9396 | 0.8161 | 0.9363 | 0.8166 | 0.9664 | 0.8978 | 0.9572 | **0.8208** | 0.9608 | 0.8306 |
| | | 2, 2 | 0.9426 | 0.6960 | 0.9364 | 0.7078 | 0.9694 | 0.7600 | 0.9553 | **0.7166** | 0.9602 | 0.7191 |
| | | 3, 3 | 0.9519 | **0.6073** | 0.9424 | 0.6237 | 0.9764 | 0.6628 | 0.9562 | 0.6339 | 0.9598 | 0.6331 |
| | | 5, 5 | 0.9621 | **0.4766** | 0.9552 | 0.4977 | 0.9823 | 0.5223 | 0.9545 | 0.5088 | 0.9583 | 0.5048 |
| | | 10, 10 | 0.9844 | **0.2918** | 0.9800 | 0.3134 | 0.9948 | 0.3240 | 0.9575 | 0.3262 | 0.9633 | 0.3203 |
| | 0.95, 0.95 | 1, 1 | 0.9400 | 0.8168 | 0.9359 | 0.8172 | 0.9656 | 0.8985 | 0.9553 | **0.8214** | 0.9600 | 0.8310 |
| | | 2, 2 | 0.9454 | 0.6975 | 0.9377 | 0.7093 | 0.9694 | 0.7611 | 0.9574 | **0.7182** | 0.9614 | 0.7205 |
| | | 3, 3 | 0.9499 | **0.6079** | 0.9444 | 0.6246 | 0.9751 | 0.6633 | 0.9535 | 0.6346 | 0.9580 | 0.6338 |
| | | 5, 5 | 0.9635 | **0.4748** | 0.9570 | 0.4959 | 0.9822 | 0.5204 | 0.9546 | 0.5075 | 0.9581 | 0.5032 |
| | | 10, 10 | 0.9813 | **0.2956** | 0.9786 | 0.3170 | 0.9911 | 0.3278 | 0.9566 | 0.3298 | 0.9604 | 0.3239 |
| 10, 10 | 0.80, 0.80 | 1, 1 | 0.9560 | 0.6576 | 0.9494 | **0.6571** | 0.9644 | 0.6857 | 0.9646 | 0.6692 | 0.9675 | 0.6678 |
| | | 2, 2 | 0.9611 | **0.5398** | 0.9534 | 0.5474 | 0.9696 | 0.5617 | 0.9661 | 0.5614 | 0.9698 | 0.5559 |
| | | 3, 3 | 0.9669 | **0.4438** | 0.9629 | 0.4548 | 0.9753 | 0.4624 | 0.9667 | 0.4694 | 0.9689 | 0.4618 |
| | | 5, 5 | 0.9792 | **0.3159** | 0.9746 | 0.3286 | 0.9864 | 0.3301 | 0.9668 | 0.3429 | 0.9696 | 0.3342 |
| | | 10, 10 | 0.9925 | **0.1599** | 0.9897 | 0.1705 | 0.9939 | 0.1684 | 0.9644 | 0.1834 | 0.9669 | 0.1762 |
| | 0.85, 0.85 | 1, 1 | 0.9591 | 0.6430 | 0.9533 | **0.6355** | 0.9660 | 0.6704 | 0.9628 | 0.6362 | 0.9674 | 0.6438 |
| | | 2, 2 | 0.9581 | **0.5391** | 0.9492 | 0.5422 | 0.9662 | 0.5602 | 0.9604 | 0.5481 | 0.9654 | 0.5490 |
| | | 3, 3 | 0.9656 | **0.4492** | 0.9604 | 0.4576 | 0.9751 | 0.4672 | 0.9619 | 0.4657 | 0.9678 | 0.4632 |
| | | 5, 5 | 0.9795 | **0.3239** | 0.9727 | 0.3359 | 0.9844 | 0.3377 | 0.9637 | 0.3457 | 0.9692 | 0.3405 |
| | | 10, 10 | 0.9912 | **0.1685** | 0.9874 | 0.1796 | 0.9942 | 0.1767 | 0.9651 | 0.1895 | 0.9712 | 0.1841 |
| | 0.90, 0.90 | 1, 1 | 0.9590 | 0.6384 | 0.9527 | 0.6286 | 0.9683 | 0.6654 | 0.9617 | **0.6264** | 0.9670 | 0.6367 |
| | | 2, 2 | 0.9610 | **0.5364** | 0.9523 | 0.5385 | 0.9699 | 0.5578 | 0.9619 | 0.5423 | 0.9677 | 0.5452 |
| | | 3, 3 | 0.9678 | **0.4497** | 0.9581 | 0.4575 | 0.9737 | 0.4677 | 0.9621 | 0.4639 | 0.9681 | 0.4629 |
| | | 5, 5 | 0.9779 | **0.3256** | 0.9700 | 0.3371 | 0.9834 | 0.3395 | 0.9604 | 0.3459 | 0.9671 | 0.3416 |
| | | 10, 10 | 0.9921 | **0.1699** | 0.9878 | 0.1809 | 0.9950 | 0.1783 | 0.9629 | 0.1903 | 0.9691 | 0.1853 |
| | 0.95, 0.95 | 1, 1 | 0.9610 | 0.6155 | 0.9520 | 0.5958 | 0.9689 | 0.6436 | 0.9563 | **0.5784** | 0.9649 | 0.6029 |
| | | 2, 2 | 0.9593 | 0.5316 | 0.9493 | 0.5274 | 0.9661 | 0.5529 | 0.9538 | **0.5214** | 0.9638 | 0.5335 |
| | | 3, 3 | 0.9610 | **0.4505** | 0.9512 | 0.4539 | 0.9712 | 0.4687 | 0.9518 | 0.4530 | 0.9628 | 0.4588 |
| | | 5, 5 | 0.9717 | **0.3324** | 0.9616 | 0.3418 | 0.9784 | 0.3462 | 0.9522 | 0.3456 | 0.9618 | 0.3458 |
| | | 10, 10 | 0.9882 | **0.1768** | 0.9803 | 0.1878 | 0.9920 | 0.1852 | 0.9512 | 0.1948 | 0.9613 | 0.1915 |
| 15, 15 | 0.80, 0.80 | 1, 1 | 0.9661 | 0.5599 | 0.9616 | **0.5517** | 0.9715 | 0.5741 | 0.9690 | 0.5537 | 0.9724 | 0.5588 |
| | | 2, 2 | 0.9634 | **0.4636** | 0.9564 | 0.4642 | 0.9691 | 0.4746 | 0.9641 | 0.4701 | 0.9683 | 0.4698 |
| | | 3, 3 | 0.9700 | **0.3772** | 0.9621 | 0.3824 | 0.9761 | 0.3867 | 0.9670 | 0.3897 | 0.9702 | 0.3868 |
| | | 5, 5 | 0.9802 | **0.2614** | 0.9742 | 0.2694 | 0.9838 | 0.2683 | 0.9670 | 0.2774 | 0.9703 | 0.2730 |
| | | 10, 10 | 0.9932 | **0.1224** | 0.9901 | 0.1295 | 0.9947 | 0.1262 | 0.9667 | 0.1367 | 0.9709 | 0.1327 |
| | 0.85, 0.85 | 1, 1 | 0.9641 | 0.5515 | 0.9577 | 0.5404 | 0.9699 | 0.5656 | 0.9651 | **0.5392** | 0.9700 | 0.5470 |
| | | 2, 2 | 0.9659 | 0.4600 | 0.9592 | **0.4593** | 0.9696 | 0.4711 | 0.9641 | 0.4629 | 0.9689 | 0.4646 |
| | | 3, 3 | 0.9731 | **0.3790** | 0.9656 | 0.3831 | 0.9772 | 0.3883 | 0.9684 | 0.3886 | 0.9729 | 0.3873 |
| | | 5, 5 | 0.9792 | **0.2646** | 0.9712 | 0.2718 | 0.9837 | 0.2717 | 0.9673 | 0.2787 | 0.9709 | 0.2753 |
| | | 10, 10 | 0.9895 | **0.1267** | 0.9863 | 0.1337 | 0.9928 | 0.1306 | 0.9635 | 0.1403 | 0.9683 | 0.1367 |

**Table 1.** *Cont.*

| $n_1, n_2$ | $\delta_1, \delta_2$ | $\sigma_1^2, \sigma_2^2$ | B.NG | | B.J | | B.U | | GCI | | FGCI | |
|---|---|---|---|---|---|---|---|---|---|---|---|---|
| | | | **CP** | **AL** | **CP** | **AL** | **CP** | **AL** | **CP** | **AL** | **CP** | **AL** |
| 15, 15 | 0.90, 0.90 | 1, 1 | 0.9687 | 0.5342 | 0.9589 | 0.5182 | 0.9718 | 0.5491 | 0.9613 | **0.5110** | 0.9684 | 0.5243 |
| | | 2, 2 | 0.9660 | 0.4561 | 0.9576 | 0.4522 | 0.9703 | 0.4677 | 0.9622 | **0.4515** | 0.9683 | 0.4572 |
| | | 3, 3 | 0.9670 | **0.3795** | 0.9584 | 0.3816 | 0.9721 | 0.3892 | 0.9594 | 0.3838 | 0.9670 | 0.3854 |
| | | 5, 5 | 0.9791 | **0.2689** | 0.9711 | 0.2752 | 0.9829 | 0.2760 | 0.9636 | 0.2801 | 0.9696 | 0.2784 |
| | | 10, 10 | 0.9923 | **0.1303** | 0.9859 | 0.1372 | 0.9939 | 0.1343 | 0.9645 | 0.1429 | 0.9700 | 0.1400 |
| | 0.95, 0.95 | 1, 1 | 0.9628 | 0.5033 | 0.9548 | 0.4812 | 0.9698 | 0.5193 | 0.9546 | **0.4658** | 0.9630 | 0.4866 |
| | | 2, 2 | 0.9606 | 0.4431 | 0.9520 | 0.4347 | 0.9676 | 0.4555 | 0.9539 | **0.4281** | 0.9638 | 0.4394 |
| | | 3, 3 | 0.9621 | 0.3755 | 0.9537 | 0.3742 | 0.9701 | 0.3858 | 0.9529 | **0.3722** | 0.9629 | 0.3782 |
| | | 5, 5 | 0.9726 | **0.2721** | 0.9632 | 0.2765 | 0.9776 | 0.2797 | 0.9532 | 0.2786 | 0.9621 | 0.2796 |
| | | 10, 10 | 0.9878 | **0.1349** | 0.9774 | 0.1414 | 0.9892 | 0.1390 | 0.9503 | 0.1457 | 0.9586 | 0.1439 |
| 30, 30 | 0.80, 0.80 | 1, 1 | 0.9775 | 0.4570 | 0.9743 | **0.4519** | 0.9783 | 0.4616 | 0.9796 | 0.4551 | 0.9806 | 0.4572 |
| | | 2, 2 | 0.9777 | **0.3618** | 0.9739 | 0.3624 | 0.9796 | 0.3656 | 0.9786 | 0.3677 | 0.9804 | 0.3665 |
| | | 3, 3 | 0.9804 | **0.2831** | 0.9764 | 0.2858 | 0.9828 | 0.2858 | 0.9805 | 0.2916 | 0.9810 | 0.2890 |
| | | 5, 5 | 0.9870 | **0.1800** | 0.9827 | 0.1838 | 0.9880 | 0.1819 | 0.9794 | 0.1892 | 0.9817 | 0.1863 |
| | | 10, 10 | 0.9952 | **0.0709** | 0.9928 | 0.0738 | 0.9955 | 0.0717 | 0.9810 | 0.0776 | 0.9841 | 0.0755 |
| | 0.85, 0.85 | 1, 1 | 0.9757 | 0.4362 | 0.9729 | 0.4276 | 0.9784 | 0.4411 | 0.9756 | **0.4270** | 0.9785 | 0.4326 |
| | | 2, 2 | 0.9716 | 0.3565 | 0.9676 | **0.3547** | 0.9739 | 0.3603 | 0.9722 | 0.3573 | 0.9767 | 0.3586 |
| | | 3, 3 | 0.9768 | **0.2840** | 0.9732 | 0.2853 | 0.9776 | 0.2871 | 0.9748 | 0.2890 | 0.9776 | 0.2883 |
| | | 5, 5 | 0.9802 | **0.1859** | 0.9777 | 0.1892 | 0.9819 | 0.1880 | 0.9727 | 0.1934 | 0.9750 | 0.1914 |
| | | 10, 10 | 0.9913 | **0.0766** | 0.9880 | 0.0796 | 0.9926 | 0.0776 | 0.9726 | 0.0829 | 0.9752 | 0.0811 |
| | 0.90, 0.90 | 1, 1 | 0.9692 | 0.4077 | 0.9642 | 0.3970 | 0.9712 | 0.4133 | 0.9649 | **0.3929** | 0.9682 | 0.4015 |
| | | 2, 2 | 0.9702 | 0.3455 | 0.9650 | 0.3421 | 0.9712 | 0.3498 | 0.9688 | **0.3420** | 0.9715 | 0.3456 |
| | | 3, 3 | 0.9690 | 0.2813 | 0.9643 | **0.2812** | 0.9713 | 0.2846 | 0.9671 | 0.2830 | 0.9713 | 0.2839 |
| | | 5, 5 | 0.9757 | **0.1899** | 0.9706 | 0.1923 | 0.9773 | 0.1922 | 0.9659 | 0.1954 | 0.9699 | 0.1944 |
| | | 10, 10 | 0.9871 | **0.0814** | 0.9847 | 0.0843 | 0.9894 | 0.0825 | 0.9656 | 0.0872 | 0.9699 | 0.0857 |
| | 0.95, 0.95 | 1, 1 | 0.9638 | 0.3574 | 0.9585 | 0.3464 | 0.9670 | 0.3632 | 0.9576 | **0.3407** | 0.9627 | 0.3497 |
| | | 2, 2 | 0.9584 | 0.3228 | 0.9520 | 0.3178 | 0.9618 | 0.3277 | 0.9542 | **0.3155** | 0.9597 | 0.3208 |
| | | 3, 3 | 0.9586 | 0.2727 | 0.9533 | 0.2714 | 0.9614 | 0.2766 | 0.9545 | **0.2710** | 0.9595 | 0.2737 |
| | | 5, 5 | 0.9676 | **0.1917** | 0.9604 | 0.1934 | 0.9706 | 0.1945 | 0.9569 | 0.1949 | 0.9628 | 0.1952 |
| | | 10, 10 | 0.9787 | **0.0877** | 0.9729 | 0.0904 | 0.9808 | 0.0890 | 0.9561 | 0.0926 | 0.9610 | 0.0916 |
| 50, 50 | 0.80, 0.80 | 1, 1 | 0.9825 | 0.3979 | 0.9805 | **0.3940** | 0.9829 | 0.3997 | 0.9836 | 0.3974 | 0.9845 | 0.3987 |
| | | 2, 2 | 0.9834 | **0.3035** | 0.9804 | 0.3037 | 0.9829 | 0.3052 | 0.9849 | 0.3082 | 0.9866 | 0.3072 |
| | | 3, 3 | 0.9860 | **0.2290** | 0.9840 | 0.2305 | 0.9861 | 0.2301 | 0.9863 | 0.2349 | 0.9873 | 0.2331 |
| | | 5, 5 | 0.9881 | **0.1369** | 0.9845 | 0.1391 | 0.9876 | 0.1376 | 0.9826 | 0.1429 | 0.9839 | 0.1408 |
| | | 10, 10 | 0.9950 | **0.0483** | 0.9935 | 0.0498 | 0.9955 | 0.0486 | 0.9827 | 0.0520 | 0.9844 | 0.0508 |
| | 0.85, 0.85 | 1, 1 | 0.9776 | 0.3628 | 0.9770 | 0.3560 | 0.9790 | 0.3652 | 0.9761 | **0.3552** | 0.9789 | 0.3603 |
| | | 2, 2 | 0.9766 | 0.2939 | 0.9745 | **0.2919** | 0.9777 | 0.2958 | 0.9773 | 0.2934 | 0.9798 | 0.2950 |
| | | 3, 3 | 0.9799 | **0.2294** | 0.9772 | 0.2296 | 0.9807 | 0.2308 | 0.9787 | 0.2320 | 0.9805 | 0.2319 |
| | | 5, 5 | 0.9830 | **0.1445** | 0.9815 | 0.1461 | 0.9850 | 0.1454 | 0.9794 | 0.1488 | 0.9812 | 0.1477 |
| | | 10, 10 | 0.9910 | **0.0546** | 0.9888 | 0.0560 | 0.9916 | 0.0549 | 0.9802 | 0.0579 | 0.9818 | 0.0570 |
| | 0.90, 0.90 | 1, 1 | 0.9700 | 0.3184 | 0.9689 | 0.3116 | 0.9728 | 0.3212 | 0.9689 | **0.3087** | 0.9707 | 0.3149 |
| | | 2, 2 | 0.9688 | 0.2746 | 0.9633 | 0.2716 | 0.9682 | 0.2766 | 0.9655 | **0.2713** | 0.9692 | 0.2743 |
| | | 3, 3 | 0.9723 | 0.2232 | 0.9702 | **0.2226** | 0.9733 | 0.2249 | 0.9712 | 0.2236 | 0.9745 | 0.2248 |
| | | 5, 5 | 0.9731 | **0.1469** | 0.9693 | 0.1479 | 0.9748 | 0.1479 | 0.9678 | 0.1497 | 0.9711 | 0.1494 |
| | | 10, 10 | 0.9824 | **0.0593** | 0.9783 | 0.0607 | 0.9832 | 0.0598 | 0.9673 | 0.0623 | 0.9701 | 0.0615 |
| | 0.95, 0.95 | 1, 1 | 0.9629 | 0.2717 | 0.9577 | 0.2667 | 0.9644 | 0.2745 | 0.9590 | **0.2650** | 0.9620 | 0.2692 |
| | | 2, 2 | 0.9618 | 0.2500 | 0.9565 | 0.2475 | 0.9633 | 0.2522 | 0.9596 | **0.2471** | 0.9621 | 0.2498 |
| | | 3, 3 | 0.9599 | 0.2118 | 0.9577 | **0.2110** | 0.9615 | 0.2137 | 0.9578 | 0.2116 | 0.9606 | 0.2130 |
| | | 5, 5 | 0.9603 | **0.1476** | 0.9578 | 0.1484 | 0.9617 | 0.1488 | 0.9528 | 0.1497 | 0.9557 | 0.1498 |
| | | 10, 10 | 0.9712 | **0.0650** | 0.9695 | 0.0663 | 0.9742 | 0.0655 | 0.9583 | 0.0676 | 0.9600 | 0.0671 |

**Table 1.** *Cont.*

| $n_1, n_2$ | $\delta_1, \delta_2$ | $\sigma_1^2, \sigma_2^2$ | B.NG | | B.J | | B.U | | GCI | | FGCI | |
|---|---|---|---|---|---|---|---|---|---|---|---|---|
| | | | CP | AL | CP | AL | CP | AL | CP | AL | CP | AL |
| 100, 100 | 0.80, 0.80 | 1, 1 | 0.9877 | 0.3289 | 0.9871 | **0.3262** | 0.9880 | 0.3298 | 0.9871 | 0.3287 | 0.9885 | 0.3300 |
| | | 2, 2 | 0.9856 | 0.2396 | 0.9846 | **0.2394** | 0.9858 | 0.2401 | 0.9874 | 0.2422 | 0.9881 | 0.2419 |
| | | 3, 3 | 0.9879 | **0.1733** | 0.9862 | 0.1739 | 0.9880 | 0.1736 | 0.9888 | 0.1765 | 0.9889 | 0.1757 |
| | | 5, 5 | 0.9917 | **0.0962** | 0.9902 | 0.0972 | 0.9909 | 0.0965 | 0.9891 | 0.0993 | 0.9901 | 0.0983 |
| | | 10, 10 | 0.9945 | **0.0293** | 0.9933 | 0.0299 | 0.9947 | 0.0294 | 0.9885 | 0.0309 | 0.9889 | 0.0304 |
| | 0.85, 0.85 | 1, 1 | 0.9809 | 0.2736 | 0.9792 | 0.2695 | 0.9813 | 0.2744 | 0.9785 | **0.2690** | 0.9795 | 0.2726 |
| | | 2, 2 | 0.9765 | 0.2205 | 0.9743 | **0.2190** | 0.9768 | 0.2212 | 0.9765 | 0.2198 | 0.9779 | 0.2212 |
| | | 3, 3 | 0.9812 | 0.1694 | 0.9795 | **0.1691** | 0.9813 | 0.1699 | 0.9805 | 0.1703 | 0.9816 | 0.1708 |
| | | 5, 5 | 0.9823 | **0.1019** | 0.9820 | 0.1025 | 0.9826 | 0.1022 | 0.9808 | 0.1038 | 0.9816 | 0.1035 |
| | | 10, 10 | 0.9870 | **0.0350** | 0.9855 | 0.0355 | 0.9870 | 0.0351 | 0.9793 | 0.0364 | 0.9800 | 0.0360 |
| | 0.90, 0.90 | 1, 1 | 0.9680 | 0.2210 | 0.9652 | 0.2184 | 0.9675 | 0.2220 | 0.9657 | **0.2181** | 0.9665 | 0.2206 |
| | | 2, 2 | 0.9646 | 0.1945 | 0.9639 | **0.1934** | 0.9662 | 0.1953 | 0.9668 | 0.1937 | 0.9684 | 0.1951 |
| | | 3, 3 | 0.9697 | 0.1587 | 0.9684 | **0.1583** | 0.9706 | 0.1592 | 0.9708 | 0.1590 | 0.9723 | 0.1597 |
| | | 5, 5 | 0.9714 | **0.1032** | 0.9700 | 0.1036 | 0.9710 | 0.1036 | 0.9690 | 0.1046 | 0.9702 | 0.1045 |
| | | 10, 10 | 0.9784 | **0.0398** | 0.9771 | 0.0403 | 0.9791 | 0.0399 | 0.9705 | 0.0410 | 0.9712 | 0.0407 |
| | 0.95, 0.95 | 1, 1 | 0.9597 | 0.1916 | 0.9577 | **0.1898** | 0.9610 | 0.1926 | 0.9588 | 0.1901 | 0.9596 | 0.1916 |
| | | 2, 2 | 0.9554 | 0.1776 | 0.9528 | **0.1767** | 0.9572 | 0.1785 | 0.9577 | 0.1775 | 0.9585 | 0.1784 |
| | | 3, 3 | 0.9586 | 0.1507 | 0.9570 | **0.1504** | 0.9587 | 0.1513 | 0.9583 | 0.1513 | 0.9594 | 0.1518 |
| | | 5, 5 | 0.9587 | **0.1040** | 0.9567 | 0.1043 | 0.9606 | 0.1044 | 0.9562 | 0.1053 | 0.9584 | 0.1053 |
| | | 10, 10 | 0.9634 | **0.0437** | 0.9616 | 0.0442 | 0.9653 | 0.0439 | 0.9552 | 0.0449 | 0.9557 | 0.0447 |
| 200, 200 | 0.80, 0.80 | 1, 1 | 0.9919 | 0.2675 | 0.9907 | **0.2655** | 0.9920 | 0.2677 | 0.9911 | 0.2673 | 0.9914 | 0.2686 |
| | | 2, 2 | 0.9872 | 0.1880 | 0.9874 | **0.1875** | 0.9886 | 0.1882 | 0.9895 | 0.1894 | 0.9893 | 0.1896 |
| | | 3, 3 | 0.9891 | **0.1309** | 0.9889 | 0.1311 | 0.9897 | 0.1310 | 0.9900 | 0.1327 | 0.9908 | 0.1324 |
| | | 5, 5 | 0.9915 | **0.0683** | 0.9912 | 0.0687 | 0.9920 | 0.0684 | 0.9913 | 0.0698 | 0.9917 | 0.0694 |
| | | 10, 10 | 0.9939 | **0.0185** | 0.9932 | 0.0187 | 0.9929 | **0.0185** | 0.9898 | 0.0192 | 0.9904 | 0.0189 |
| | 0.85, 0.85 | 1, 1 | 0.9804 | 0.1929 | 0.9790 | **0.1913** | 0.9810 | 0.1931 | 0.9801 | 0.1916 | 0.9807 | 0.1932 |
| | | 2, 2 | 0.9771 | 0.1575 | 0.9766 | **0.1568** | 0.9786 | 0.1577 | 0.9780 | 0.1574 | 0.9785 | 0.1582 |
| | | 3, 3 | 0.9796 | 0.1212 | 0.9787 | **0.1209** | 0.9808 | 0.1213 | 0.9804 | 0.1218 | 0.9811 | 0.1221 |
| | | 5, 5 | 0.9825 | **0.0720** | 0.9809 | 0.0722 | 0.9818 | 0.0721 | 0.9816 | 0.0729 | 0.9823 | 0.0728 |
| | | 10, 10 | 0.9868 | **0.0234** | 0.9851 | 0.0236 | 0.9851 | 0.0235 | 0.9808 | 0.0240 | 0.9812 | 0.0239 |
| | 0.90, 0.90 | 1, 1 | 0.9686 | 0.1552 | 0.9684 | **0.1544** | 0.9678 | 0.1556 | 0.9684 | 0.1549 | 0.9704 | 0.1558 |
| | | 2, 2 | 0.9635 | 0.1377 | 0.9650 | **0.1372** | 0.9651 | 0.1380 | 0.9657 | 0.1381 | 0.9669 | 0.1386 |
| | | 3, 3 | 0.9679 | 0.1123 | 0.9669 | **0.1122** | 0.9683 | 0.1125 | 0.9684 | 0.1129 | 0.9693 | 0.1132 |
| | | 5, 5 | 0.9670 | **0.0727** | 0.9659 | 0.0728 | 0.9670 | **0.0727** | 0.9676 | 0.0735 | 0.9670 | 0.0734 |
| | | 10, 10 | 0.9754 | **0.0271** | 0.9736 | 0.0273 | 0.9758 | 0.0272 | 0.9691 | 0.0277 | 0.9707 | 0.0276 |
| | 0.95, 0.95 | 1, 1 | 0.9582 | 0.1347 | 0.9568 | **0.1341** | 0.9572 | 0.1350 | 0.9588 | 0.1348 | 0.9593 | 0.1353 |
| | | 2, 2 | 0.9540 | 0.1258 | 0.9516 | **0.1255** | 0.9557 | 0.1261 | 0.9549 | 0.1263 | 0.9556 | 0.1266 |
| | | 3, 3 | 0.9561 | 0.1068 | 0.9561 | **0.1067** | 0.9581 | 0.1071 | 0.9586 | 0.1075 | 0.9599 | 0.1077 |
| | | 5, 5 | 0.9598 | **0.0734** | 0.9575 | 0.0735 | 0.9592 | 0.0736 | 0.9582 | 0.0742 | 0.9598 | 0.0742 |
| | | 10, 10 | 0.9608 | **0.0301** | 0.9611 | 0.0302 | 0.9619 | **0.0301** | 0.9574 | 0.0306 | 0.9585 | 0.0305 |

Note: The values shown in bold represent the shortest expected lengths.

**Table 2.** The coverage probabilities and average lengths of two-sided 95% HPD and confidence intervals for the difference between CQVs in delta-lognormal distributions are presented for cases with unequal sample sizes.

| $n_1, n_2$ | $\delta_1, \delta_2$ | $\sigma_1^2, \sigma_2^2$ | B.NG | | B.J | | B.U | | GCI | | FGCI | |
|---|---|---|---|---|---|---|---|---|---|---|---|---|
| | | | CP | AL | CP | AL | CP | AL | CP | AL | CP | AL |
| 5, 15 | 0.80, 0.80 | 1, 1 | 0.9457 | 0.7145 | 0.9407 | 0.7179 | 0.9696 | 0.7704 | 0.9627 | 0.7342 | 0.9654 | **0.7325** |
| | | 2, 2 | 0.9518 | **0.5935** | 0.9474 | 0.6040 | 0.9733 | 0.6371 | 0.9631 | 0.6220 | 0.9649 | 0.6156 |
| | | 3, 3 | 0.9598 | **0.4955** | 0.9556 | 0.5094 | 0.9790 | 0.5325 | 0.9655 | 0.5281 | 0.9674 | 0.5194 |
| | | 5, 5 | 0.9692 | **0.3713** | 0.9633 | 0.3872 | 0.9842 | 0.4015 | 0.9596 | 0.4060 | 0.9602 | 0.3967 |
| | | 10, 10 | 0.9872 | **0.2124** | 0.9848 | 0.2265 | 0.9940 | 0.2330 | 0.9648 | 0.2459 | 0.9669 | 0.2374 |
| | 0.85, 0.85 | 1, 1 | 0.9476 | 0.6946 | 0.9433 | 0.6916 | 0.9699 | 0.7443 | 0.9591 | **0.6960** | 0.9630 | 0.7027 |
| | | 2, 2 | 0.9529 | **0.5857** | 0.9473 | 0.5934 | 0.9705 | 0.6235 | 0.9597 | 0.6016 | 0.9637 | 0.6020 |
| | | 3, 3 | 0.9608 | **0.4962** | 0.9531 | 0.5086 | 0.9761 | 0.5283 | 0.9628 | 0.5187 | 0.9659 | 0.5160 |
| | | 5, 5 | 0.9711 | **0.3691** | 0.9624 | 0.3850 | 0.9858 | 0.3949 | 0.9616 | 0.3967 | 0.9667 | 0.3920 |
| | | 10, 10 | 0.9857 | **0.2089** | 0.9819 | 0.2242 | 0.9919 | 0.2261 | 0.9595 | 0.2377 | 0.9643 | 0.2324 |
| | 0.90, 0.90 | 1, 1 | 0.9454 | 0.6838 | 0.9412 | 0.6779 | 0.9639 | 0.7333 | 0.9569 | **0.6783** | 0.9617 | 0.6884 |
| | | 2, 2 | 0.9463 | 0.5833 | 0.9396 | 0.5889 | 0.9648 | 0.6213 | 0.9552 | **0.5949** | 0.9610 | 0.5980 |
| | | 3, 3 | 0.9529 | **0.4960** | 0.9472 | 0.5070 | 0.9705 | 0.5281 | 0.9576 | 0.5152 | 0.9638 | 0.5146 |
| | | 5, 5 | 0.9667 | **0.3740** | 0.9585 | 0.3890 | 0.9793 | 0.3997 | 0.9575 | 0.3996 | 0.9626 | 0.3960 |
| | | 10, 10 | 0.9850 | **0.2151** | 0.9795 | 0.2304 | 0.9922 | 0.2326 | 0.9586 | 0.2433 | 0.9630 | 0.2385 |
| | 0.95, 0.95 | 1, 1 | 0.9377 | 0.6672 | 0.9345 | 0.6590 | 0.9580 | 0.7163 | 0.9523 | **0.6560** | 0.9577 | 0.6689 |
| | | 2, 2 | 0.9373 | 0.5754 | 0.9315 | 0.5796 | 0.9582 | 0.6135 | 0.9488 | 0.5830 | 0.9548 | **0.5882** |
| | | 3, 3 | 0.9439 | 0.4956 | 0.9410 | 0.5052 | 0.9626 | 0.5279 | 0.9520 | **0.5114** | 0.9569 | 0.5129 |
| | | 5, 5 | 0.9571 | **0.3767** | 0.9488 | 0.3907 | 0.9698 | 0.4025 | 0.9506 | 0.4002 | 0.9572 | 0.3979 |
| | | 10, 10 | 0.9775 | **0.2175** | 0.9699 | 0.2323 | 0.9871 | 0.2349 | 0.9509 | 0.2446 | 0.9561 | 0.2403 |
| 10, 50 | 0.80, 0.80 | 1, 1 | 0.9653 | 0.5416 | 0.9572 | **0.5408** | 0.9721 | 0.5581 | 0.9664 | 0.5496 | 0.9727 | 0.5490 |
| | | 2, 2 | 0.9666 | **0.4292** | 0.9596 | 0.4346 | 0.9722 | 0.4414 | 0.9667 | 0.4461 | 0.9705 | 0.4416 |
| | | 3, 3 | 0.9721 | **0.3418** | 0.9648 | 0.3495 | 0.9772 | 0.3515 | 0.9680 | 0.3622 | 0.9716 | 0.3565 |
| | | 5, 5 | 0.9827 | **0.2283** | 0.9759 | 0.2370 | 0.9859 | 0.2355 | 0.9649 | 0.2506 | 0.9706 | 0.2444 |
| | | 10, 10 | 0.9946 | **0.1034** | 0.9922 | 0.1099 | 0.9960 | 0.1073 | 0.9685 | 0.1224 | 0.9715 | 0.1177 |
| | 0.85, 0.85 | 1, 1 | 0.9679 | 0.5177 | 0.9633 | **0.5110** | 0.9750 | 0.5339 | 0.9694 | 0.5115 | 0.9751 | 0.5178 |
| | | 2, 2 | 0.9655 | **0.4273** | 0.9579 | 0.4297 | 0.9723 | 0.4396 | 0.9654 | 0.4352 | 0.9716 | 0.4356 |
| | | 3, 3 | 0.9684 | **0.3479** | 0.9608 | 0.3538 | 0.9737 | 0.3577 | 0.9636 | 0.3615 | 0.9704 | 0.3596 |
| | | 5, 5 | 0.9794 | **0.2395** | 0.9731 | 0.2480 | 0.9834 | 0.2468 | 0.9652 | 0.2578 | 0.9725 | 0.2541 |
| | | 10, 10 | 0.9906 | **0.1110** | 0.9889 | 0.1182 | 0.9926 | 0.1150 | 0.9668 | 0.1284 | 0.9728 | 0.1247 |
| | 0.90, 0.90 | 1, 1 | 0.9517 | 0.4969 | 0.9511 | 0.4890 | 0.9601 | 0.5140 | 0.9587 | **0.4870** | 0.9649 | 0.4956 |
| | | 2, 2 | 0.9540 | **0.4190** | 0.9510 | 0.4203 | 0.9612 | 0.4313 | 0.9628 | 0.4240 | 0.9679 | 0.4260 |
| | | 3, 3 | 0.9611 | **0.3458** | 0.9581 | 0.3511 | 0.9663 | 0.3558 | 0.9649 | 0.3574 | 0.9709 | 0.3568 |
| | | 5, 5 | 0.9652 | **0.2404** | 0.9642 | 0.2483 | 0.9728 | 0.2476 | 0.9630 | 0.2573 | 0.9690 | 0.2543 |
| | | 10, 10 | 0.9812 | **0.1145** | 0.9810 | 0.1216 | 0.9844 | 0.1185 | 0.9655 | 0.1312 | 0.9699 | 0.1278 |
| | 0.95, 0.95 | 1, 1 | 0.9500 | 0.4688 | 0.9478 | 0.4546 | 0.9569 | 0.4866 | 0.9540 | **0.4456** | 0.9604 | 0.4621 |
| | | 2, 2 | 0.9466 | 0.4079 | 0.9478 | 0.4058 | 0.9548 | 0.4209 | 0.9573 | **0.4032** | 0.9643 | 0.4109 |
| | | 3, 3 | 0.9513 | **0.3433** | 0.9486 | 0.3465 | 0.9578 | 0.3537 | 0.9588 | 0.3482 | 0.9634 | 0.3515 |
| | | 5, 5 | 0.9529 | **0.2462** | 0.9524 | 0.2533 | 0.9594 | 0.2536 | 0.9542 | 0.2590 | 0.9608 | 0.2585 |
| | | 10, 10 | 0.9681 | **0.1225** | 0.9685 | 0.1298 | 0.9713 | 0.1266 | 0.9545 | 0.1378 | 0.9616 | 0.1355 |
| 15, 100 | 0.80, 0.80 | 1, 1 | 0.9563 | **0.4561** | 0.9416 | 0.4503 | 0.9611 | 0.4644 | 0.9450 | 0.4528 | 0.9546 | **0.4561** |
| | | 2, 2 | 0.9595 | **0.3624** | 0.9431 | 0.3636 | 0.9653 | 0.3688 | 0.9450 | 0.3688 | 0.9571 | 0.3684 |
| | | 3, 3 | 0.9638 | **0.2852** | 0.9503 | 0.2894 | 0.9702 | 0.2904 | 0.9451 | 0.2965 | 0.9557 | 0.2943 |
| | | 5, 5 | 0.9774 | **0.1861** | 0.9670 | 0.1917 | 0.9818 | 0.1895 | 0.9434 | 0.2005 | 0.9556 | 0.1973 |
| | | 10, 10 | 0.9933 | **0.0764** | 0.9865 | 0.0809 | 0.9951 | 0.0780 | 0.9422 | 0.0892 | 0.9542 | 0.0866 |
| | 0.85, 0.85 | 1, 1 | 0.9647 | 0.4307 | 0.9514 | 0.4229 | 0.9683 | 0.4395 | 0.9564 | **0.4221** | 0.9656 | 0.4282 |
| | | 2, 2 | 0.9663 | 0.3557 | 0.9539 | **0.3556** | 0.9696 | 0.3624 | 0.9580 | 0.3587 | 0.9663 | 0.3600 |
| | | 3, 3 | 0.9685 | **0.2856** | 0.9591 | 0.2890 | 0.9716 | 0.2909 | 0.9587 | 0.2944 | 0.9655 | 0.2934 |
| | | 5, 5 | 0.9776 | **0.1907** | 0.9694 | 0.1960 | 0.9793 | 0.1943 | 0.9612 | 0.2037 | 0.9686 | 0.2012 |
| | | 10, 10 | 0.9886 | **0.0816** | 0.9857 | 0.0861 | 0.9908 | 0.0833 | 0.9590 | 0.0937 | 0.9683 | 0.0914 |

**Table 2.** *Cont.*

| $n_1, n_2$ | $\delta_1, \delta_2$ | $\sigma_1^2, \sigma_2^2$ | B.NG | | B.J | | B.U | | GCI | | FGCI | |
|---|---|---|---|---|---|---|---|---|---|---|---|---|
| | | | CP | AL | CP | AL | CP | AL | CP | AL | CP | AL |
| 15, 100 | 0.90, 0.90 | 1, 1 | 0.9634 | 0.4041 | 0.9567 | 0.3935 | 0.9672 | 0.4136 | 0.9627 | **0.3900** | 0.9691 | 0.3992 |
| | | 2, 2 | 0.9591 | 0.3457 | 0.9522 | **0.3436** | 0.9645 | 0.3528 | 0.9581 | 0.3440 | 0.9664 | 0.3477 |
| | | 3, 3 | 0.9617 | **0.2847** | 0.9570 | 0.2869 | 0.9671 | 0.2903 | 0.9601 | 0.2899 | 0.9696 | 0.2909 |
| | | 5, 5 | 0.9697 | **0.1933** | 0.9651 | 0.1982 | 0.9731 | 0.1970 | 0.9610 | 0.2041 | 0.9683 | 0.2028 |
| | | 10, 10 | 0.9804 | **0.0864** | 0.9779 | 0.0911 | 0.9817 | 0.0883 | 0.9604 | 0.0978 | 0.9683 | 0.0958 |
| | 0.95, 0.95 | 1, 1 | 0.9529 | 0.3756 | 0.9479 | 0.3610 | 0.9587 | 0.3860 | 0.9495 | **0.3532** | 0.9591 | 0.3669 |
| | | 2, 2 | 0.9493 | 0.3346 | 0.9444 | 0.3295 | 0.9538 | 0.3425 | 0.9526 | **0.3256** | 0.9604 | 0.3331 |
| | | 3, 3 | 0.9504 | 0.2806 | 0.9463 | 0.2806 | 0.9570 | 0.2868 | 0.9514 | **0.2803** | 0.9605 | 0.2840 |
| | | 5, 5 | 0.9560 | **0.1964** | 0.9533 | 0.2004 | 0.9595 | 0.2006 | 0.9544 | 0.2041 | 0.9630 | 0.2045 |
| | | 10, 10 | 0.9677 | **0.0921** | 0.9677 | 0.0966 | 0.9698 | 0.0941 | 0.9566 | 0.1023 | 0.9625 | 0.1009 |
| 50, 200 | 0.80, 0.80 | 1, 1 | 0.9843 | 0.3442 | 0.9818 | **0.3422** | 0.9860 | 0.3455 | 0.9827 | 0.3462 | 0.9860 | 0.3463 |
| | | 2, 2 | 0.9834 | **0.2513** | 0.9804 | 0.2519 | 0.9834 | 0.2522 | 0.9820 | 0.2562 | 0.9845 | 0.2549 |
| | | 3, 3 | 0.9848 | **0.1823** | 0.9824 | 0.1837 | 0.9860 | 0.1829 | 0.9818 | 0.1879 | 0.9841 | 0.1864 |
| | | 5, 5 | 0.9879 | **0.1023** | 0.9844 | 0.1040 | 0.9882 | 0.1027 | 0.9803 | 0.1078 | 0.9835 | 0.1063 |
| | | 10, 10 | 0.9934 | **0.0321** | 0.9923 | 0.0331 | 0.9946 | 0.0323 | 0.9812 | 0.0354 | 0.9836 | 0.0346 |
| | 0.85, 0.85 | 1, 1 | 0.9791 | 0.2905 | 0.9764 | **0.2860** | 0.9791 | 0.2920 | 0.9760 | 0.2867 | 0.9805 | 0.2903 |
| | | 2, 2 | 0.9762 | 0.2361 | 0.9723 | **0.2348** | 0.9769 | 0.2374 | 0.9745 | 0.2362 | 0.9772 | 0.2373 |
| | | 3, 3 | 0.9764 | **0.1821** | 0.9736 | 0.1823 | 0.9762 | 0.1829 | 0.9747 | 0.1843 | 0.9782 | 0.1844 |
| | | 5, 5 | 0.9798 | **0.1117** | 0.9775 | 0.1128 | 0.9804 | 0.1121 | 0.9747 | 0.1154 | 0.9775 | 0.1147 |
| | | 10, 10 | 0.9867 | **0.0395** | 0.9851 | 0.0405 | 0.9870 | 0.0397 | 0.9731 | 0.0425 | 0.9769 | 0.0419 |
| | 0.90, 0.90 | 1, 1 | 0.9678 | 0.2488 | 0.9657 | 0.2442 | 0.9693 | 0.2506 | 0.9658 | **0.2436** | 0.9690 | 0.2478 |
| | | 2, 2 | 0.9613 | 0.2168 | 0.9606 | **0.2147** | 0.9634 | 0.2180 | 0.9641 | 0.2148 | 0.9660 | 0.2169 |
| | | 3, 3 | 0.9639 | 0.1760 | 0.9620 | **0.1756** | 0.9649 | 0.1770 | 0.9644 | 0.1764 | 0.9673 | 0.1773 |
| | | 5, 5 | 0.9721 | **0.1145** | 0.9708 | 0.1153 | 0.9716 | 0.1152 | 0.9696 | 0.1170 | 0.9722 | 0.1169 |
| | | 10, 10 | 0.9730 | **0.0443** | 0.9733 | 0.0452 | 0.9737 | 0.0446 | 0.9672 | 0.0469 | 0.9704 | 0.0465 |
| | 0.95, 0.95 | 1, 1 | 0.9588 | 0.2137 | 0.9553 | 0.2104 | 0.9616 | 0.2155 | 0.9586 | **0.2099** | 0.9619 | 0.2127 |
| | | 2, 2 | 0.9553 | 0.1976 | 0.9502 | **0.1959** | 0.9563 | 0.1992 | 0.9551 | 0.1961 | 0.9583 | 0.1978 |
| | | 3, 3 | 0.9545 | 0.1672 | 0.9522 | **0.1666** | 0.9550 | 0.1684 | 0.9540 | 0.1674 | 0.9562 | 0.1682 |
| | | 5, 5 | 0.9600 | **0.1157** | 0.9580 | 0.1163 | 0.9604 | 0.1165 | 0.9585 | 0.1176 | 0.9610 | 0.1177 |
| | | 10, 10 | 0.9639 | **0.0489** | 0.9616 | 0.0498 | 0.9645 | 0.0493 | 0.9562 | 0.0512 | 0.9591 | 0.0509 |
| 100, 200 | 0.80, 0.80 | 1, 1 | 0.9890 | 0.3039 | 0.9889 | **0.3016** | 0.9892 | 0.3041 | 0.9892 | 0.3042 | 0.9895 | 0.3052 |
| | | 2, 2 | 0.9877 | 0.2159 | 0.9873 | **0.2157** | 0.9878 | 0.2163 | 0.9881 | 0.2184 | 0.9891 | 0.2181 |
| | | 3, 3 | 0.9845 | **0.1529** | 0.9843 | 0.1534 | 0.9859 | 0.1531 | 0.9846 | 0.1558 | 0.9865 | 0.1551 |
| | | 5, 5 | 0.9890 | **0.0822** | 0.9886 | 0.0829 | 0.9894 | 0.0823 | 0.9864 | 0.0848 | 0.9884 | 0.0841 |
| | | 10, 10 | 0.9944 | **0.0234** | 0.9932 | 0.0238 | 0.9941 | **0.0234** | 0.9882 | 0.0247 | 0.9893 | 0.0243 |
| | 0.85, 0.85 | 1, 1 | 0.9832 | 0.2362 | 0.9802 | **0.2333** | 0.9824 | 0.2370 | 0.9802 | 0.2337 | 0.9816 | 0.2363 |
| | | 2, 2 | 0.9788 | 0.1917 | 0.9780 | **0.1906** | 0.9793 | 0.1922 | 0.9798 | 0.1912 | 0.9806 | 0.1925 |
| | | 3, 3 | 0.9817 | 0.1472 | 0.9793 | **0.1469** | 0.9824 | 0.1475 | 0.9814 | 0.1480 | 0.9822 | 0.1484 |
| | | 5, 5 | 0.9813 | **0.0878** | 0.9792 | 0.0882 | 0.9812 | 0.0880 | 0.9795 | 0.0894 | 0.9816 | 0.0892 |
| | | 10, 10 | 0.9866 | **0.0293** | 0.9859 | 0.0297 | 0.9862 | 0.0294 | 0.9795 | 0.0305 | 0.9802 | 0.0302 |
| | 0.90, 0.90 | 1, 1 | 0.9674 | 0.1906 | 0.9662 | **0.1889** | 0.9674 | 0.1914 | 0.9666 | 0.1890 | 0.9680 | 0.1907 |
| | | 2, 2 | 0.9708 | 0.1685 | 0.9689 | **0.1676** | 0.9713 | 0.1691 | 0.9710 | 0.1682 | 0.9717 | 0.1693 |
| | | 3, 3 | 0.9685 | 0.1374 | 0.9667 | **0.1371** | 0.9677 | 0.1378 | 0.9681 | 0.1379 | 0.9700 | 0.1384 |
| | | 5, 5 | 0.9689 | **0.0890** | 0.9668 | 0.0893 | 0.9706 | 0.0893 | 0.9675 | 0.0902 | 0.9685 | 0.0902 |
| | | 10, 10 | 0.9747 | **0.0338** | 0.9741 | 0.0341 | 0.9756 | **0.0338** | 0.9698 | 0.0348 | 0.9707 | 0.0346 |
| | 0.95, 0.95 | 1, 1 | 0.9596 | 0.1655 | 0.9578 | **0.1643** | 0.9605 | 0.1662 | 0.9601 | 0.1648 | 0.9623 | 0.1658 |
| | | 2, 2 | 0.9575 | 0.1539 | 0.9555 | **0.1532** | 0.9578 | 0.1545 | 0.9595 | 0.1540 | 0.9598 | 0.1547 |
| | | 3, 3 | 0.9559 | 0.1306 | 0.9528 | **0.1304** | 0.9557 | 0.1311 | 0.9566 | 0.1312 | 0.9568 | 0.1316 |
| | | 5, 5 | 0.9566 | **0.0898** | 0.9556 | 0.0900 | 0.9566 | 0.0901 | 0.9565 | 0.0909 | 0.9577 | 0.0909 |
| | | 10, 10 | 0.9650 | **0.0373** | 0.9629 | 0.0377 | 0.9629 | 0.0374 | 0.9580 | 0.0383 | 0.9602 | 0.0381 |

Note: The values shown in bold represent the shortest expected lengths.

In Figures 3 and 4, the Bayesian method based on Jeffrey's prior exhibits good performance regarding coverage probabilities for cases with small variances. In contrast, the coverage probabilities of GCI are significantly lower than those of other methods for cases with large variances. Furthermore, both the GCI and FGCI methods consistently show similar coverage probabilities. Additionally, the average lengths of intervals show similar results across all methods. Furthermore, it is worth noting that in Figures 1–4, the average lengths tend to narrow for all methods as the variances or sample sizes increase.

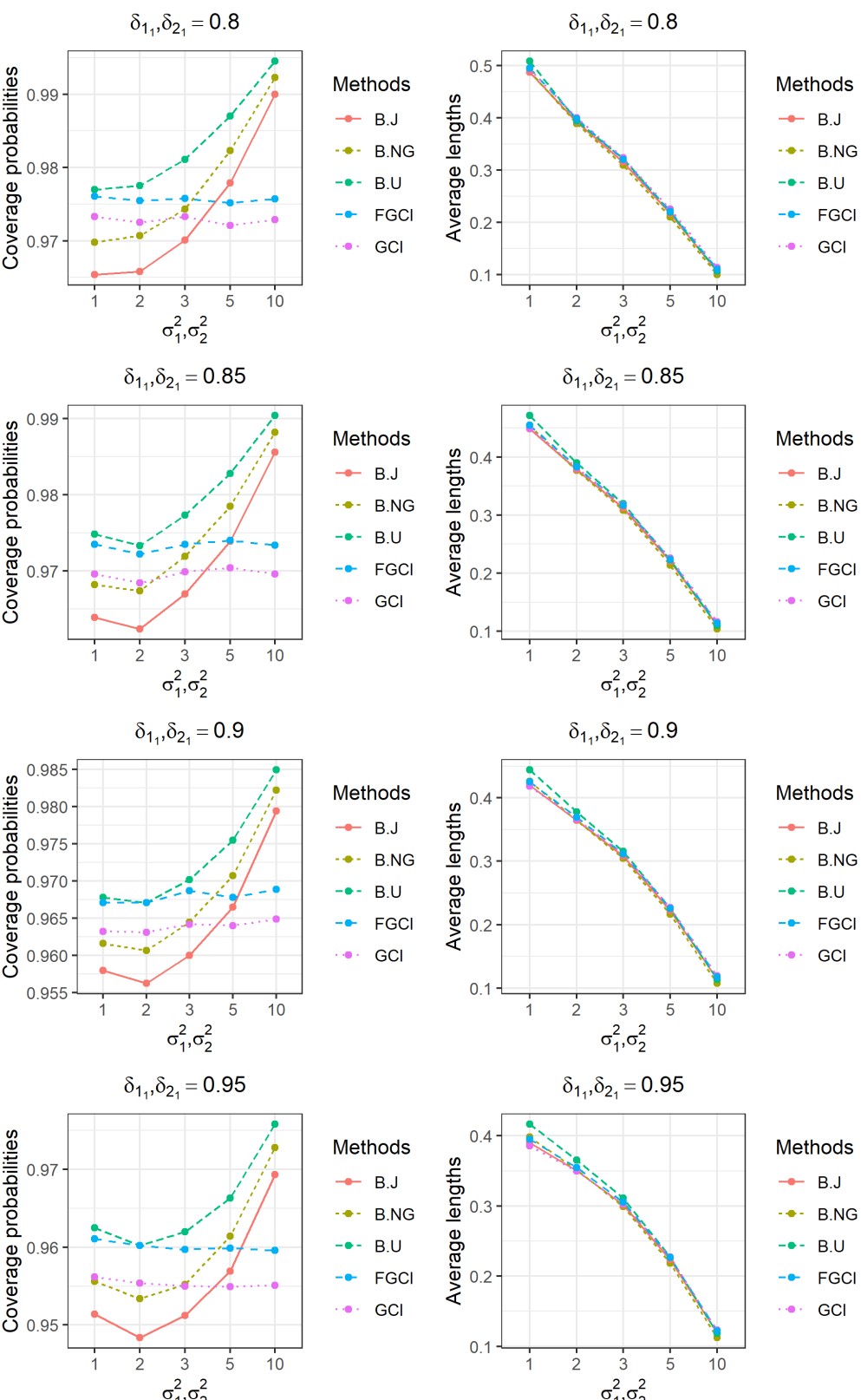

**Figure 3.** Comparison of the coverage probabilities and average lengths of proposed methods for cases with varying variances and probabilities of positive values.

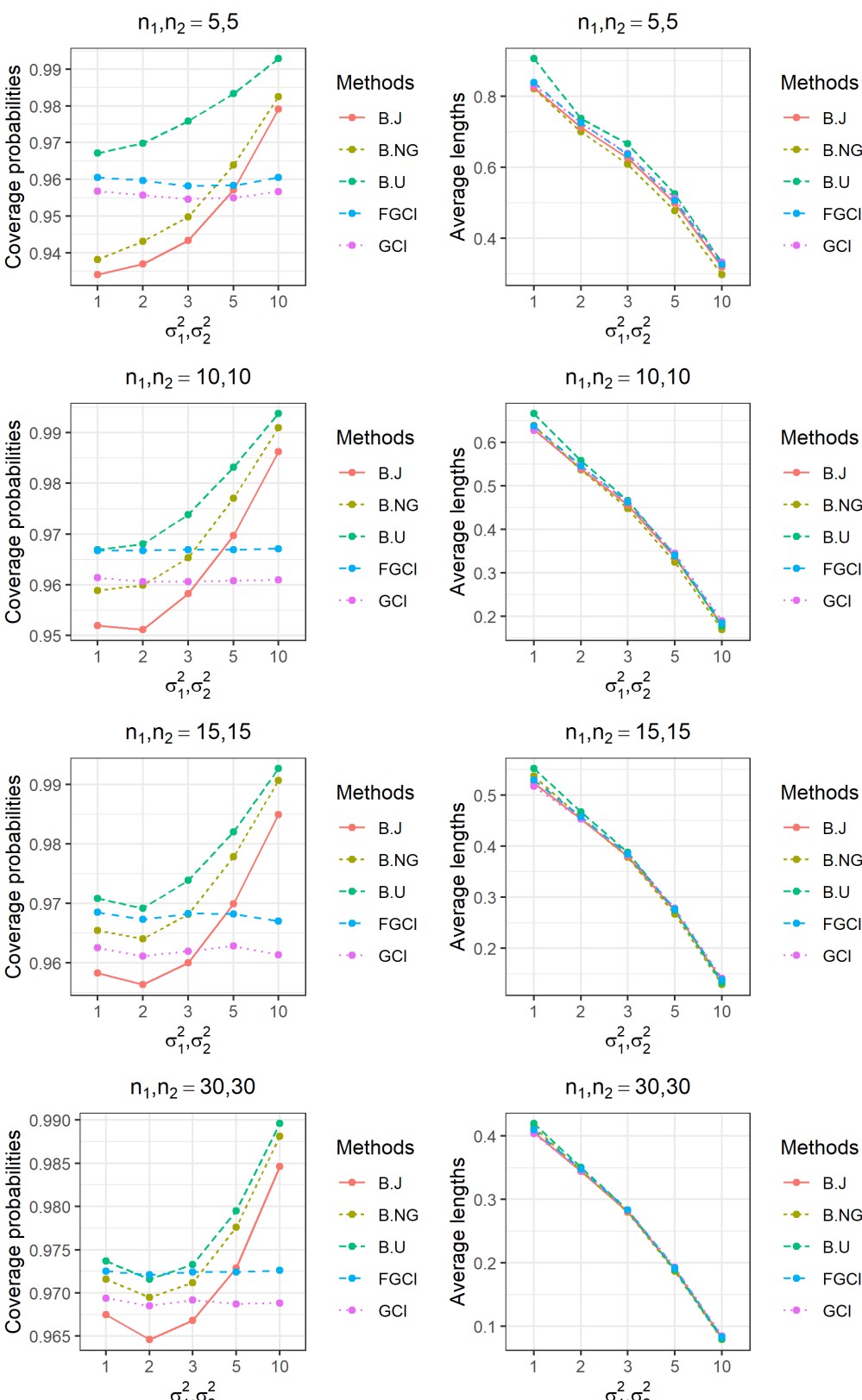

**Figure 4.** *Cont.*

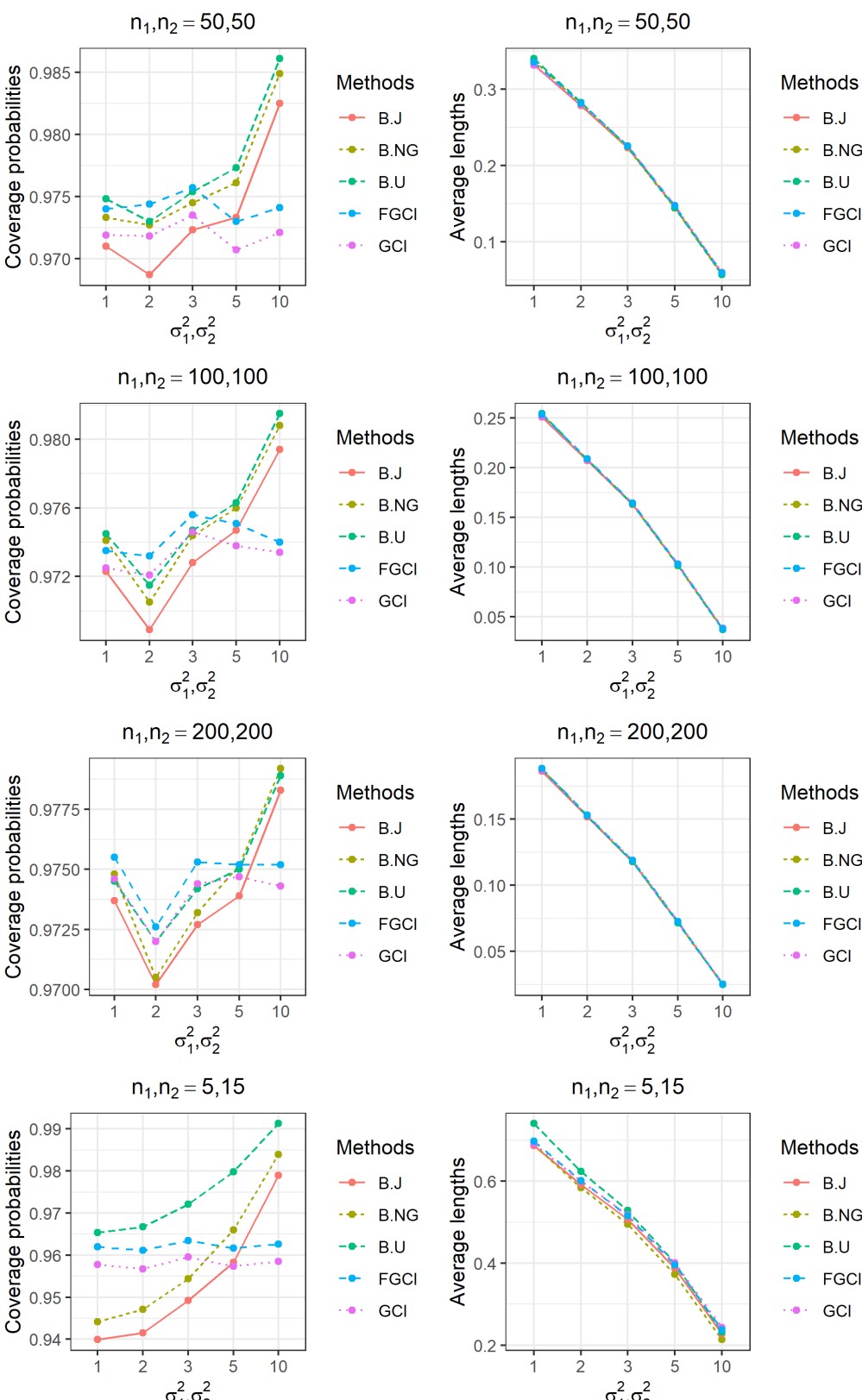

**Figure 4.** *Cont.*

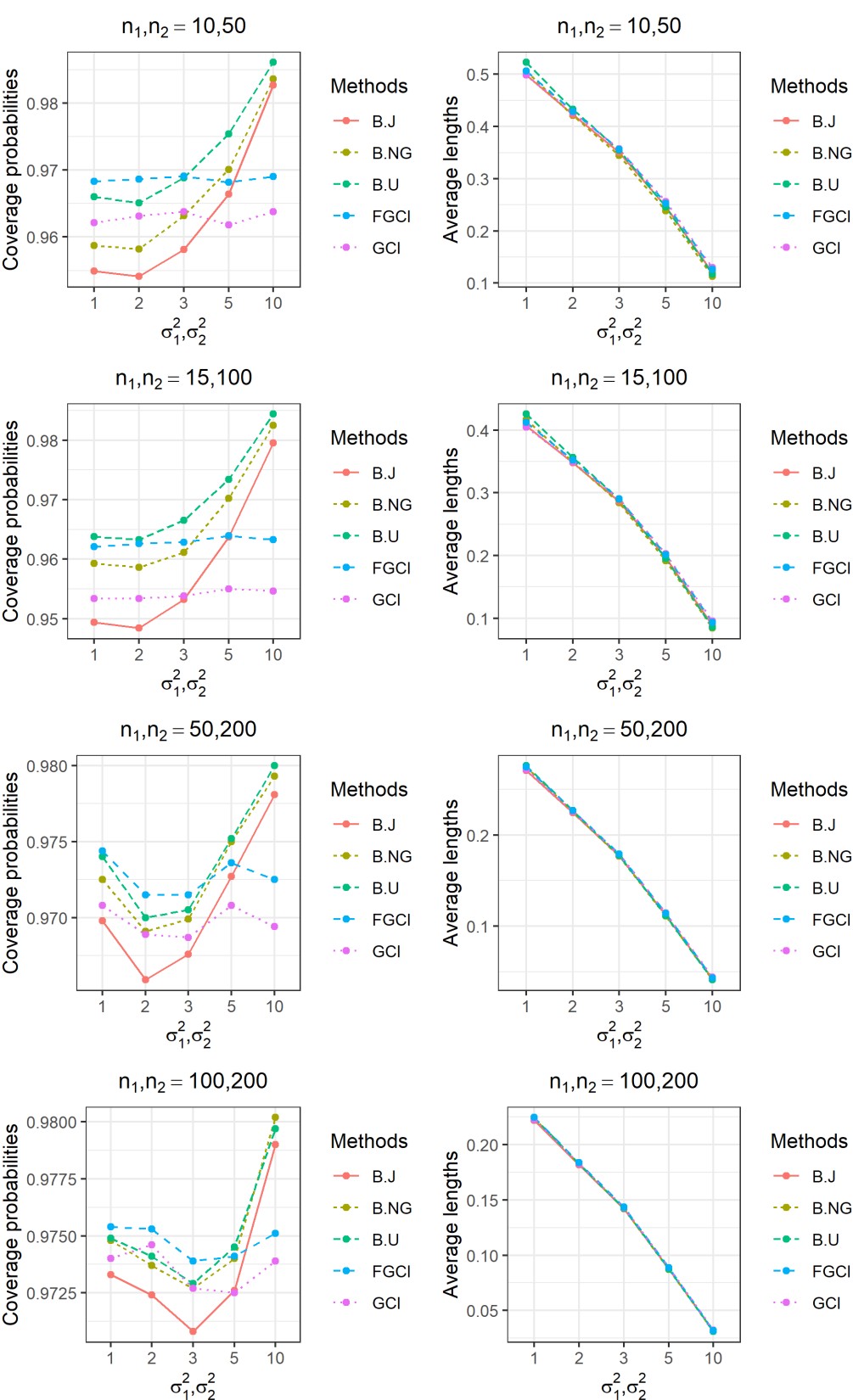

**Figure 4.** Comparison of coverage probabilities and average lengths of proposed methods for cases with varying variances and sample sizes.

*3.2. Empirical Study*

The rainfall data collected from Bang Klam and Singhanakhon, Songkhla, Thailand, between 2018 and 2022 conform to a delta-lognormal distribution. These data, consisting of positive and zero values, are particularly interesting for the study. The Southern-East Coast Meteorological Center provided the monthly rainfall data presented in Table 3. Both datasets include positive and zero values, with the proportion of positive rainfall data in Bang Klam and Singhanakhon being 53 out of 60 and 48 out of 60, respectively. Descriptive statistics were calculated for the data from Bang Klam and Singhanakhon, resulting in the following values: sample sizes $(n_1, n_2) = (60, 60)$, proportions of positive values $(\hat{\delta}_{1_1}, \hat{\delta}_{2_1}) = (0.88, 0.80)$, means $(\hat{\mu}_1, \hat{\mu}_2) = (89.3717, 137.5783)$, and variances $(\hat{\sigma}_1^2, \hat{\sigma}_2^2) = (18{,}551.1790, 38{,}863.9817)$.

The positive values in both rainfall datasets exhibited right-skewness, as shown in Figure 5. The minimum Akaike information criteria (AIC) and the Bayesian information criteria (BIC) were utilized to confirm the distribution of the data. The results in Table 4 demonstrate that both datasets have lower AIC and BIC values for the lognormal distribution, conforming their adherence to this distribution. Additionally, density and normal Q-Q plots of the log-transformed rainfall data, as depicted in Figure 6, further validate their lognormal distribution.

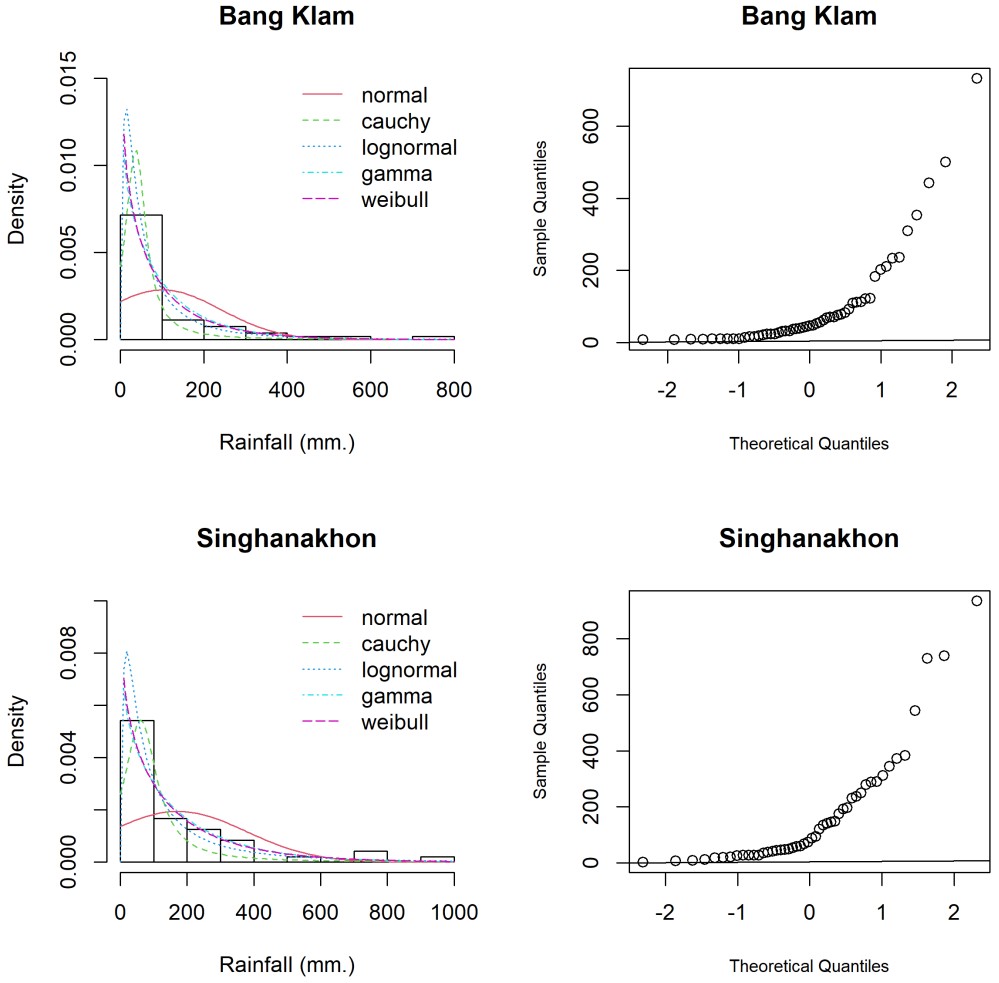

**Figure 5.** Density and normal Q-Q plots of monthly rainfall data in Bang Klam and Singhanakhon, Songkhla, Thailand (2018–2022).

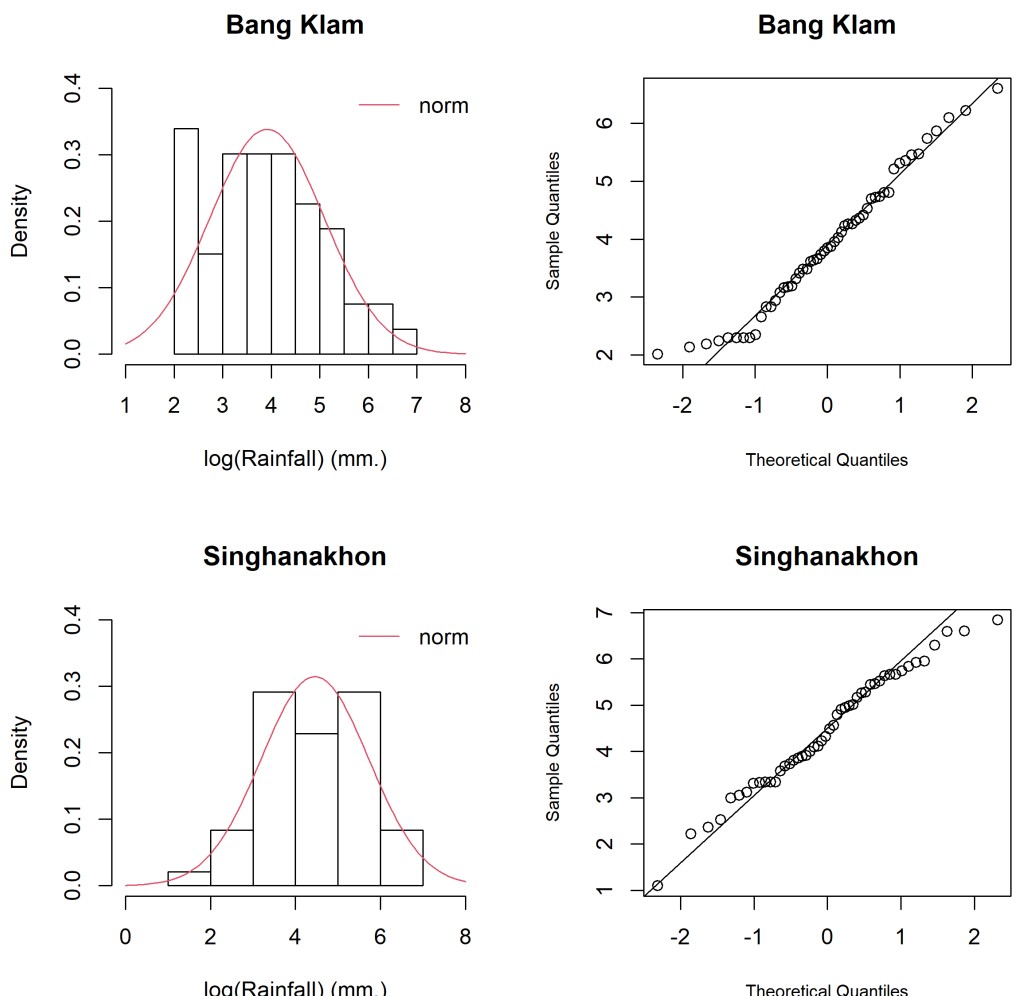

**Figure 6.** Density and normal Q-Q plots of log-transformed monthly rainfall data in Bang Klam and Singhanakhon, Songkhla, Thailand (2018–2022).

**Table 3.** Summary of monthly rainfall data for Bang Klam and Singhanakhon, Songkhla, Thailand (2018–2022).

| Month | Bang Klam | | | | | Singhanakhon | | | | |
|---|---|---|---|---|---|---|---|---|---|---|
| | **2022** | **2021** | **2020** | **2019** | **2018** | **2022** | **2021** | **2020** | **2019** | **2018** |
| Jan. | 69 | 39 | 0 | 113.8 | 71.1 | 121.3 | 49.4 | 0 | 146.6 | 237.5 |
| Feb. | 211 | 0 | 24 | 0 | 0 | 193.1 | 0 | 50.2 | 0 | 0 |
| Mar. | 14.3 | 19 | 0 | 0 | 10 | 141.2 | 47.5 | 0 | 0 | 9.2 |
| Apr. | 24.4 | 10 | 17 | 30.5 | 27.5 | 290.4 | 21.2 | 0 | 0 | 45 |
| May | 82.5 | 112.4 | 9.5 | 0 | 47 | 250 | 28.2 | 0 | 12.5 | 39.8 |
| Jun. | 75.8 | 78.3 | 23.7 | 10.5 | 48 | 69 | 289.3 | 3 | 0 | 22.6 |
| Jul. | 32.7 | 52.5 | 93 | 56 | 7.5 | 75.7 | 175.6 | 28 | 89 | 0 |
| Aug. | 42 | 8.5 | 10 | 9 | 32.5 | 96 | 60.2 | 35.8 | 10.7 | 28.3 |
| Sept. | 21.7 | 10 | 37 | 44.5 | 17 | 61.6 | 0 | 20 | 28.4 | 41.9 |
| Oct. | 202.2 | 38 | 71.1 | 122.4 | 183 | 313 | 27.6 | 135.3 | 196.9 | 373.3 |
| Nov. | 353.6 | 500.6 | 442.8 | 236.9 | 109.5 | 344.5 | 738.2 | 729.7 | 150.2 | 383.7 |
| Dec. | 732.5 | 122.1 | 233.8 | 62 | 309.6 | 933.6 | 232 | 543.9 | 54.7 | 279.9 |

**Table 4.** AIC and BIC values for monthly rainfall data in Bang Klam and Singhanakorn, Songkhla, Thailand (2018–2022).

| Areas | Criteria | Distributions | | | | |
|---|---|---|---|---|---|---|
| | | **Normal** | **Cauchy** | **Lognormal** | **Gamma** | **Weibull** |
| Bang Klam | AIC | 677.8336 | 627.4157 | **586.7384** | 598.2644 | 596.5048 |
| | BIC | 681.7742 | 631.3562 | **590.6790** | 602.2050 | 600.4454 |
| Singhanakorn | AIC | 651.0329 | 632.7343 | **590.8083** | 593.3023 | 592.6424 |
| | BIC | 654.7753 | 636.4767 | **594.5507** | 597.0447 | 596.3848 |

Note: The values shown in bold represent the minimum AIC and BIC.

Table 5 displays the 95% HPD and confidence intervals for the difference between CQVs of the monthly rainfall data from Bang Klam and Singhanakhon. The interval lengths of all methods are similar, consistent with the findings of the simulation study. However, it should be noted that the HPD and confidence intervals generated by all methods exhibit conflicting lower and upper bounds, indicating that these intervals contain zero values. Therefore, there is no significant difference in the monthly rainfall dispersion between both areas.

**Table 5.** The 95% HPD and confidence intervals for the difference in CQVs of monthly rainfall data between Bang Klam and Singhanakorn, Songkhla, Thailand.

| Methods | Lower Bound | Upper Bound | Length |
|---|---|---|---|
| B.NG | −0.2489 | 0.0591 | 0.3080 |
| B.J | −0.2472 | 0.0775 | 0.3247 |
| B.U | −0.2620 | 0.0545 | 0.3165 |
| GCI | −0.2542 | 0.0661 | 0.3203 |
| FGCI | −0.2560 | 0.0743 | 0.3303 |

## 4. Discussion and Conclusions

This study aimed to construct HPD and confidence intervals for the difference between CQVs of delta-lognormal distributions. We proposed the Bayesian method based on three priors: normal gamma, Jeffreys, and uniform priors, along with the GCI and FGCI methods. To evaluate the performance of these methods, we assessed their coverage probabilities and average lengths under various simulation scenarios.

The findings indicate that the Bayesian approach based on Jeffreys' prior is suitable for cases with slight variances or small sample sizes. Conversely, as the variance increases, the GCI method outperforms the others. However, it is essential to note that this study focuses on quartiles and specifically applies when the probabilities of positive values are 0.8 or higher. Furthermore, we computed the HPD and confidence intervals for the CQVs of rainfall data from two areas, which followed a delta-lognormal distribution. The empirical results align with the findings from the simulation study, demonstrating that the interval lengths are similar across all methods. In conclusion, we recommend using the Bayesian approach based on Jeffreys' prior and the GCI method to construct HPD and confidence intervals for the difference between CQVs of delta-lognormal distributions.

**Author Contributions:** Conceptualization, N.Y. and S.-A.N.; methodology, N.Y. and S.-A.N.; software, N.Y.; validation, N.Y. and S.-A.N.; formal analysis, N.Y. and S.-A.N.; investigation, S.-A.N.; resources, N.Y.; data curation, N.Y.; writing—original draft preparation, N.Y.; writing—review and editing, N.Y. and S.-A.N.; visualization, N.Y.; supervision, S.-A.N.; project administration, S.-A.N.; funding acquisition, N.Y. All authors have read and agreed to the published version of the manuscript.

**Funding:** This work (Grant No. RGNS 64-131) was financially supported by Office of the Permanent Secretary, Ministry of Higher Education, Science, Research and Innovation, Thailand Science Research and Innovation.

**Institutional Review Board Statement:** Not applicable.

**Informed Consent Statement:** Not applicable.

**Data Availability Statement:** The rainfall data from Bang Klam and Singhanakhon, Songkhla, Thailand, were collected by the Southern-East Coast Meteorological Center.

**Acknowledgments:** This work (Grant No. RGNS 64-131) was supported by Office of the Permanent Secretary, Ministry of Higher Education, Science, Research and Innovation (OPS MHESI), Thailand Science Research and Innovation (TSRI) and University of Phayao.

**Conflicts of Interest:** The authors declare no conflict of interest.

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
