# Peer review of "Bayesian Estimation for the Difference between Coefficients of Quartile Variation of Delta-Lognormal Distributions: An Application to Rainfall in Thailand"

_symmetry, doi:10.3390/sym15071383_

Round 1

Reviewer 1 Report

This paper investigated the construction of confidence intervals for difference of CQVs. One class of methods investigated are Bayesian methods with different priors. In addition to Bayesian methods, they also investigated the application of generalized confidence. Extensive simulations are conducted and interesting phenomenons have been found. I have the following questions for the authors. Why the coverage probability decreases as sample size increases for some methods shown in Figure 1? Is it because the average length is dressing? If so, is there a trade-off between CP and AL? 

I think the paper is readable and I can easily understand the key methods.

Reviewer 3 Report

1. In the Introduction, the explanations of coefficients of quartile variation need to be strengthened. Additionally, the role of the lognormal distribution in climate change studies should be elaborated upon. Furthermore, the review of statistical methods should be improved by incorporating recent advances in Bayesian deep learning methods, such as those discussed in “10.1016/j.ress.2023.109181”, and generalized inference "10.1016/j.ress.2021.108136". It is also important to address the limitations and advantages of different interval estimate methods.

2. In Section 2, it is recommended to include discussions on how to choose hyperparameters in the normal-gamma prior. Explaining the process of determining the values of these hyperparameters would be beneficial. Moreover, when priors are chosen in forms other than the normal-gamma distribution, the posterior distribution may not have closed-form solutions or may not follow regular distributions. Therefore, it is necessary to discuss alternative inference methods such as variational Bayes “10.1109/TR.2023.3263940” and Markov chain Monte Carlo (MCMC) sampling.

3. In Section 3, it is suggested that the authors include cases with small sample sizes to assess the performance of the proposed methods. For instance, considering sample sizes of (n1,n2)=(5,5) and (10,10) would provide insights into how the two methods perform under such conditions.

Round 2

Reviewer 2 Report

All of my comments have been addressed. I suggest considering the manuscript for publication in its present form. 

Reviewer 3 Report

The authors have revised the manuscript well.